# Engineering artificial photosynthesis based on rhodopsin for $CO_2$ fixation

Weiming Tu [1], Jiabao Xu [1], Ian P. Thompson [1] & Wei E. Huang [1]✉

Microbial rhodopsin, a significant contributor to sustaining life through light harvesting, holds untapped potential for carbon fixation. Here, we construct an artificial photosynthesis system which combines the proton-pumping ability of rhodopsin with an extracellular electron uptake mechanism, establishing a pathway to drive photoelectrosynthetic $CO_2$ fixation by *Ralstonia eutropha* (also known as *Cupriavidus necator*) H16, a facultatively chemolithoautotrophic soil bacterium. *R. eutropha* is engineered to heterologously express an extracellular electron transfer pathway of *Shewanella oneidensis* MR-1 and *Gloeobacter* rhodopsin (GR). Employing GR and the outer-membrane conduit MtrCAB from *S. oneidensis*, extracellular electrons and GR-driven proton motive force are integrated into *R. eutropha*'s native electron transport chain (ETC). Inspired by natural photosynthesis, the photoelectrochemical system splits water to supply electrons to *R. eutropha* via the Mtr outer-membrane route. The light-activated proton pump - GR, supported by canthaxanthin as an antenna, powers ATP synthesis and reverses the ETC to regenerate NADH/NADPH, facilitating *R. eutropha*'s biomass synthesis from $CO_2$. Overexpression of a carbonic anhydrase further enhances $CO_2$ fixation. This artificial photosynthesis system has the potential to advance the development of efficient photosynthesis, redefining our understanding of the ecological role of microbial rhodopsins in nature.

Microbial carbon dioxide ($CO_2$) fixation through photosynthesis is one of the foundations of the global carbon cycle[1,2]. Photosynthetic microbes harvest solar radiation to convert $CO_2$ and water into organic compounds, contributing around 50% of primary productivity on earth[3]. In nature, almost all known light-harvesting mechanisms in microorganisms are either chlorophyll- or rhodopsin-based systems[4]. Chlorophyll-based photosystems are multi-component pigment-protein complexes (i.e., photosystem I and II) coupled with a series of redox protein complexes to form a photosynthetic electron transport chain, which use light as the energy source to split water, accompanied by the production of ATP and reductants (e.g., NADPH). In comparison, rhodopsin-based photosystems are much simpler with only a light-activated proton pump generating proton motive force[5]. There was a lot of evidence suggesting that rhodopsin can support ATP synthesis to enhance inorganic carbon assimilation in the presence of organic electron donors[6,7]. However, rhodopsin cannot solely drive the $CO_2$ fixation pathway with its proton pumping, due to the lack of electron generation. Microbial rhodopsins as the most widespread proteins in the microbial world are hypothesised to have a significant role in microbial carbon fixation[8]. Recently, rhodopsin phototrophy has been shown to generate sufficient energy for reversing electron transfer which can drive NADH synthesis via NADH dehydrogenase[9] and then power $CO_2$ fixation[10]. These findings indicated the proton gradient generated by rhodopsin can support the regeneration of NAD(P)H via an electron transport chain, providing a driving force for $CO_2$ reduction[10].

Microbial rhodopsins are widely used as optogenetic tools in synthetic biology and have been successfully expressed in model bacteria such as *Escherichia coli*[11], *Ralstonia eutropha*[10] and *Shewanella oneidensis*[9]. Numerous studies[12-14] demonstrated that the engineered bacteria with rhodopsin can convert light energy into intracellular

[1]Department of Engineering Science, University of Oxford, Oxford OX1 3PJ, UK. ✉e-mail: wei.huang@eng.ox.ac.uk

chemical energy. For example, the engineered *E. coli* with rhodopsin has been shown to improve bioproduction in the light[12]. Additionally, the expression of rhodopsin in the electroactive bacterium *S. oneidensis* can power the electrosynthesis of reduced products in cathodic conditions[9]. Integrating a rhodopsin-based photosystem with chemoautotrophic bacteria represents a promising approach to increase biosynthesis from $CO_2$. Chemoautotrophic bacteria can utilise inorganic chemicals or an electrode as the electron donor to assimilate $CO_2$ into valuable compounds. *R. eutropha* H16, a chemolithoautotroph, is a model microbial chassis, owing to its $CO_2$ fixation pathway and metabolic versatility. We recently engineered the autotrophic bacterium *R. eutropha* H16 with a *Gloeobacter* rhodopsin (GR) and created a redox loop by integrating it with an external electrode[10]. Electrons supplied by the electrode were transferred into the engineered *R. eutropha* to drive $CO_2$ fixation. The process was mediated by an electron-shuttling molecule such as flavin, and powered by rhodopsins[10]. While this system can effectively incorporate $CO_2$, it displayed a relatively low electron transfer rate and efficiency. We hypothesised that engineering an efficient electron transfer interface on the cell membrane could be a key strategy for improving the electron transport from the electrode to the cells. Natural electroactive bacteria, such as *S. oneidensis* MR-1, use the outer-membrane-bound Mtr pathway, facilitating bidirectional electron transfer between extracellular substrates and the quinone pool[15,16]. In the model photosynthetic bacterium *Rhodopseudomonas palustris* TIE-1, there are also cytochromes presented on the outer membrane to enable the electron transfer between intracellular and extracellular electron acceptors and electrons[17]. It is hypothesised that artificially combining extracellular electron uptake with intracellular reverse electron transfer could lead to the construction of a synthetic photosynthetic electron transport chain in *R. eutropha* H16. In addition, the proton-pumping capacity could be another limiting factor to rhodopsin-based photoelectrosynthesis, which is associated with ATP and NADH synthesis[18]. As with *R. palustris*, the proton gradient not only

drives ATP production via ATP synthase but also plays a key role in reversing the function of proton-translocating NADH dehydrogenase for regeneration of reducing equivalents[17]. Therefore, increasing the proton-pumping activity of rhodopsin may also increase the efficiency of the artificial system.

In this work, *R. eutropha* H16 is engineered to establish a photon and electron harvesting system by the heterologous expression of the outer-membrane conduit MtrCAB complex from *S. oneidensis* MR-1 and the GR protein from *Gloeobacter violaceus*. This synthetic biology design combining the Mtr pathway and rhodopsin-based phototrophy enables the construction of an electrochemically driven photosynthetic electron transport chain. The engineered *R. eutropha* can directly take electrons from a cathode through the Mtr pathway and use the proton motive force generated by light-activated rhodopsin to power ATP and NAD(P)H synthesis. We integrate the photoelectrosynthetic *R. eutropha* with a solar-driven electrochemical system to achieve artificial photosynthesis (Fig. 1). To enhance $CO_2$ fixation efficiency, flavin and canthaxanthin are introduced to combine with Mtr and GR, respectively, improving the electron transfer rate and proton-pumping capacity. Additionally, a β-carbonic anhydrase (*can*) is overexpressed to enhance $CO_2$ fixation, as increased *can* levels facilitate the interconversion between bicarbonate and $CO_2$. Overall, artificial photosynthesis could help design alternative methods for $CO_2$ fixation, providing a valuable reference for engineering non-photoelectrosynthetic bacteria into photoelectrotrophs, and offering insights into future investigations of the potential interplay between the rhodopsin-based metabolism and extracellular electron transfer, along with their joint effect on microbial $CO_2$ fixation.

## Results

### Engineering a synthetic Mtr-mediated electron transport chain

A plasmid containing a gene pathway encoding MtrCAB (pLO11a-MtrCAB) was created under the control of an arabinose-inducible $P_{BAD}$

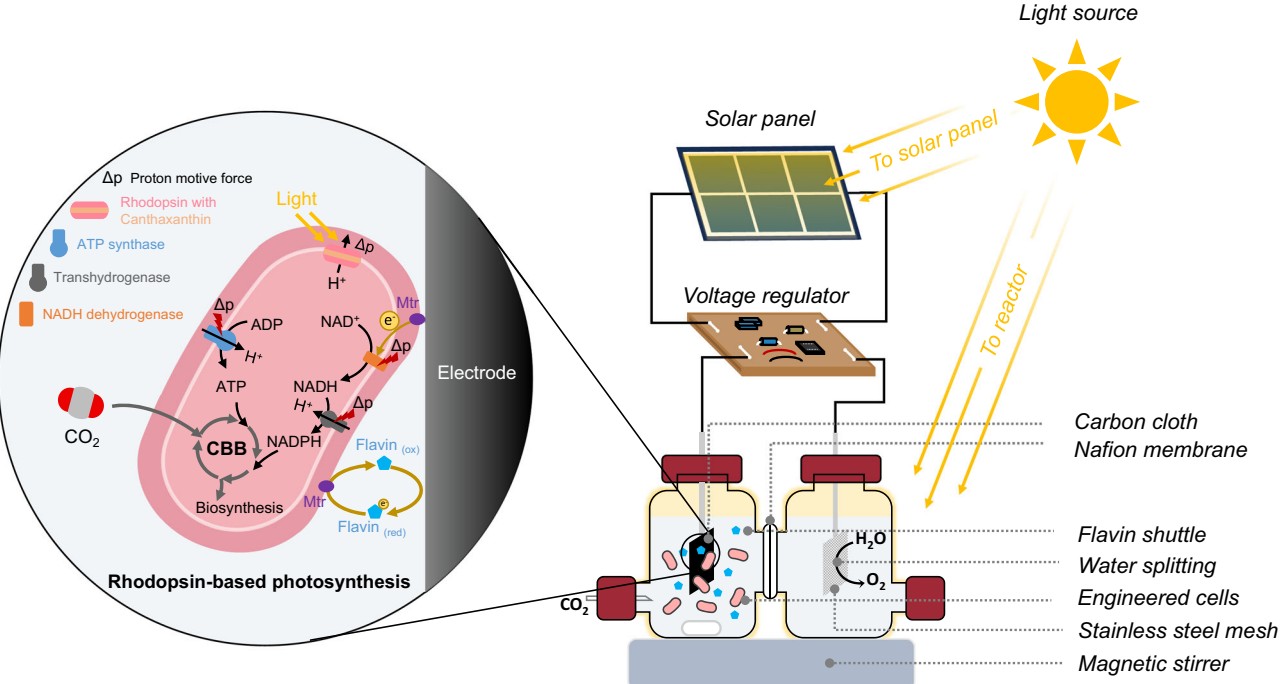

**Fig. 1 | Construction of an artificial photosynthesis system by integrating a photoelectrochemical system with genetically engineered cells expressing rhodopsin and an outer-membrane conduit MtrCAB.** In the artificial photosynthesis system, light energy is absorbed by a solar panel and rhodopsin to generate electricity and drive the metabolism, respectively. The engineered bacteria obtain electrons from the electrode, mediated by the Mtr complex and flavins, to synthesise reducing power (i.e., NADH and NADPH). In the presence of ATP and NADPH, the Calvin–Benson–Bassham (CBB) cycle is activated to drive $CO_2$ fixation for biosynthesis.

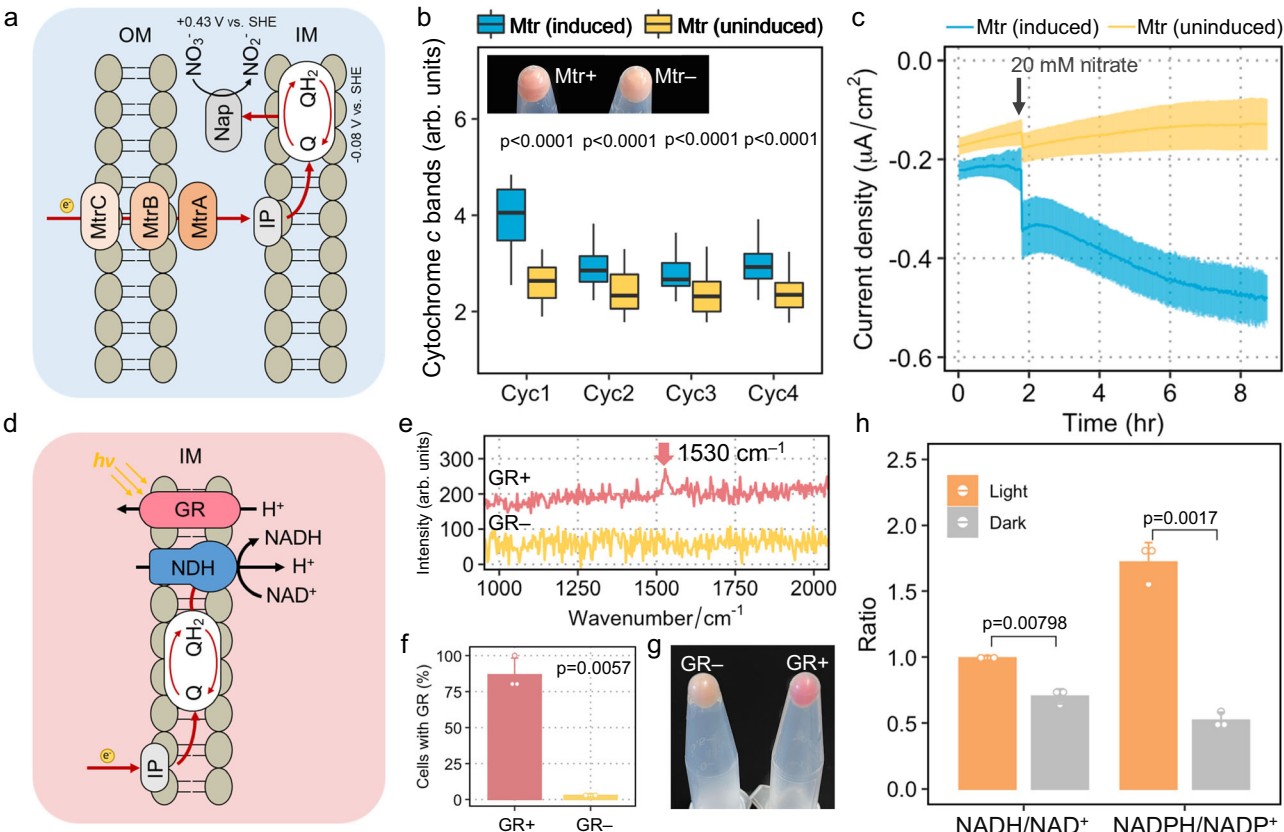

**Fig. 2 | Construction of a photosynthetic electron transport chain. a** The MtrCAB complex mediates inward electron transfer to reduce nitrate (OM outer membrane, IM inner membrane, IP inner-membrane proteins, Nap nitrate reductases, Q quinones). **b** Semi-quantification of *c*-type cytochrome levels in cells with or without Mtr complexes at the single-cell level by Raman analysis. Cyc1, Cyc2, Cyc3 and Cyc4 represent the quantification of four bands at 748, 1128, 1312 and 1584 cm$^{-1}$. The number of measured cells = 234. The lower and upper quartiles are drawn as lines outside the box. The rectangle in the box plots represents the second and third quartiles with a line inside representing the median. The whiskers indicated the minima and maxima. **c** Current consumption by induced and uninduced *R. eutropha*-Mtr with the addition of 20 mM nitrate (data show means ± SD of three biological replicates). **d** Light-activated GR drives NADH dehydrogenase (NDH) in reverse to synthesise NADH from quinone. **e** A typical Raman spectrum of a cell with GR complexes identified by a band at ~1530 cm$^{-1}$ (red) and a typical Raman spectrum with the band intensity below background noise identified as a cell without GR (yellow). **f** The percentage of GR-expressing cells, i.e., with the 1530 cm$^{-1}$ GR band, in induced (number of measured cells $n$ = 297) and uninduced (number of measured cells $n$ = 306) groups. Data show means ± SD of three biological replicates. **g** Uninduced and arabinose-induced cell pellets were yellow and pink, respectively. **h** NADH/NAD$^+$ and NADPH/NADP$^+$ ratios in the engineered cells under light and dark conditions (data show means ± SD of three biological replicates). Statistics were performed with a two-sided Student's *t*-test (exact *p*-values are provided in the display). Source data are provided as a Source Data file.

promoter. MtrCAB is a multi-heme protein complex linking the intracellular electron transport chain with extracellular substrates (Fig. 2a). Cytochrome *c* maturation is required for heme insertion[19]. Unlike *E. coli* which cannot express cytochrome *c* aerobically[19], *R. eutropha* H16 can synthesise cytochromes under aerobic conditions, the same as that of electroactive *Shewanella oneidensis* MR-1 cells (Supplementary Fig. 1). The plasmid pLO11a-MtrCAB was transferred into *R. eutropha* H16 to create *R. eutropha*-Mtr (Supplementary Table 1). After induction by arabinose, the cell pellet of induced *R. eutropha*-Mtr showed a red colour (insets in Fig. 2b), compared with that of uninduced cells. The red colour was attributed to the expression of MtrCAB complexes on the cell membrane which is consistent with the results reported in *E. coli*[19]. Single-cell Raman analysis revealed that Raman spectra of cells expressing Mtr displayed a significant increase in bands associated with the cytochromes (Fig. 2b and Supplementary Fig. 2). This is in good agreement with a previous study reporting that *S. oneidensis* MR-1's cytochromes levels decreased due to the deletion of Mtr genes[20]. MtrC and MtrA are *c*-type cytochromes; thus, the elevated cytochromes suggest the synthesis of Mtr in *R. eutropha*-Mtr. Proteins from whole cell extracts were separated by sodium dodecyl-sulfate polyacrylamide gel electrophoresis (Supplementary Method 1), and heme *c*

containing proteins MtrC and MtrA were identified by heme straining (Supplementary Fig. 3). Collectively, the results confirm the expression of the Mtr pathway in *R. eutropha*.

Furthermore, electrochemical tests were performed to confirm the functions of Mtr in *R. eutropha*. Since *R. eutropha* is able to carry out a nitrate-reducing metabolism[21], we hypothesised that in the presence of nitrate as an electron acceptor, the Mtr pathway in *R. eutropha*-Mtr could obtain cathodic electrons for denitrification via native nitrate reductases (Fig. 2a). The precultured *R. eutropha*-Mtr with and without arabinose induction were incubated in potentiostatic-controlled bioreactors under cathodic conditions ($-500$ mV$_{Ag/AgCl}$). After a period of acclimation, 20 mM nitrate was added to the cathodic chamber. We observed that the addition of nitrate accelerated *R. eutropha*-Mtr consuming current with a significant current drop over 6 h, while little current change was detected in the control group with the Mtr-uninduced strain (Fig. 2c). The observation indicates that electron flux from a cathode to nitrate reductases was due to the Mtr complex in *R. eutropha*-Mtr. The nitrate respiration pathway is part of the electron transport chain; thus, the expression of the Mtr complex creates an electrical connection between cellular energy metabolism and the electrode.

## Combination of the electron transport chain with a rhodopsin-based photosystem

Reducing power such as NADH and NADPH is the key driving force to power $CO_2$ fixation. Although the Mtr-mediated electron transfer system can support electron flow inwardly, there is an energy hurdle over intracellular electron transfer. We expect that electrons flow from a cathode to Mtr ($-300\ mV_{SHE}$ to $+100\ mV_{SHE}$)[16], then to the quinones pool ($-80\ mV_{SHE}$ for menaquinone) mediated by inner-membrane proteins (IPs) and onto NADH dehydrogenase for the generation of NADH ($-320\ mV_{SHE}$). However, the electron transfer from quinol to $NAD^+$ is thermodynamically unfavourable[9]. Previous studies have demonstrated that introducing a rhodopsin-based proton pump can drive the process by providing extra energy in the form of proton motive force[9,10]. Therefore, in this study, we combined the Mtr pathway with a gene encoding *Gloeobacter* rhodopsin (GR) to form a phototrophic extracellular electron uptake pathway (Fig. 2d). The GR-based photosystem can harvest light energy around 530 nm[8] and act as a light-activated proton pump. Previous in vitro studies have demonstrated that holo-GR was able to generate a proton motive force under light and had a two-fold faster turnover rate than other proteorhodopsins (PRs)[22]. Hence, a plasmid containing genes encoding both MtrCAB and GR (pLO11a-MtrCAB-GR) was transferred to *R. eutropha* to create *R. eutropha*-GR-Mtr (Supplementary Table 1).

As a consequence of GR expression, arabinose-induced cells in the cell pellet were pink in colour, compared with that of uninduced bacteria (Fig. 2g). The pink colour was attributed to the presence of GR-retinal complexes on the cell membrane. Single-cell Raman analysis revealed that Raman spectra of cells expressing GR displayed a characteristic rhodopsin band at 1530 $cm^{-1}$ (Fig. 2e), consistent with previous report[10]. The GR complexes were detected in 87% of the induced cells by examining single-cell Raman spectra of 270 individual cells, compared to 3% in the uninduced cells (number of measured cells $n = 306$) (Fig. 2f). After establishing the GR-based photosystem with the Mtr pathway, the *R. eutropha*-GR-Mtr strain was employed to investigate the possibility that the extracellular electrons could drive the reducing power generation in *R. eutropha* under light, by incubating the strain in cathodic conditions with an electrode as the electron source. After 2-day incubation, reducing power accumulated in light conditions due to the energy provided by the functional GR. The NADH and NADPH levels were significantly increased in *R. eutropha*-GR-Mtr under the light compared to that in the dark (Fig. 2h). Interestingly, in the light, the NADPH/$NADP^+$ ratio was higher than NADH/

$NAD^+$ (Fig. 2h), indicating that the strain could be more inclined to take extracellular electrons for driving NADPH-dependent reactions such as $CO_2$ fixation.

## Light-driven metabolism with $CO_2$ as the sole carbon source

It is reasonable to assume that the reducing power NADPH would energise the $CO_2$ fixation pathway (Calvin cycle) in *R. eutropha* H16. Biosynthesis requires a catalyst, energy and building blocks (such as $CO_2$ and $H_2O$). *R. eutropha*-GR-Mtr was used as a catalyst to harvest light energy for $CO_2$ fixation. Heavy water $D_2O$ has been employed to probe general cellular metabolic activities[23]. We hypothesised that the photoelectro-autotrophic *R. eutropha*-GR-Mtr, in the presence of both $CO_2$ and reducing power NADH/NADPH, can incorporate $D_2O$ into cell metabolisms, including the regeneration of NAD(P)H and the synthesis of 3-phosphoglycerate (3PG) (Supplementary Fig. 4a). Thus, in the presence of $D_2O$, carbon−deuterium (C−D) bond will form only in metabolically active cells, which can be readily detected by single-cell Raman analysis (Supplementary Fig. 4a).

The precultured *R. eutropha*-GR-Mtr was inoculated into cathode chambers which were operated in four conditions: dark and no $CO_2$, dark with $CO_2$, light and no $CO_2$, and light with $CO_2$. After 2-day incubation, C−D bands originated from newly formed C−D bonds were only observed in the group with light and $CO_2$ in both averaged Raman spectra (Fig. 3a) and single-cell analysis (Fig. 3b; number of measured individual cells = 294), suggesting that *R. eutropha*-GR-Mtr was active to perform light-driven $CO_2$ fixation. In the case of *R. eutropha*-GR, no significant C−D bands were observed in the presence of light and $CO_2$ (Supplementary Fig. 5), indicating the importance of Mtr in extracellular electron transfer. In particular, the single band at 1003 $cm^{-1}$ is characteristic of the phenyl ring of phenylalanine (an aromatic amino acid), which becomes three bands at 987, 975, and 961 $cm^{-1}$ due to the utilisation of $D_2O$ in *R. europha*-GR-Mtr (Supplementary Fig. 4b). These isotopic shifts in Raman spectra are in good agreement with the phenylalanine deuteration reported in previous work[24]. In addition, the display of a band at 1050 $cm^{-1}$ in Raman spectra from the cells in the dark indicates bicarbonate presence, since the continuous supply of $CO_2$ without consumption would result in the accumulation of bicarbonate intracellularly in *R. eutropha*-GR-Mtr. These results indicate that the introduction of MtrCBA and GR in *R. eutropha*-GR-Mtr leads to biosynthesis from $CO_2$. In summary, these observations confirm that *R. eutropha*-GR-Mtr has been converted into a photoelectro-autotrophic bacterium.

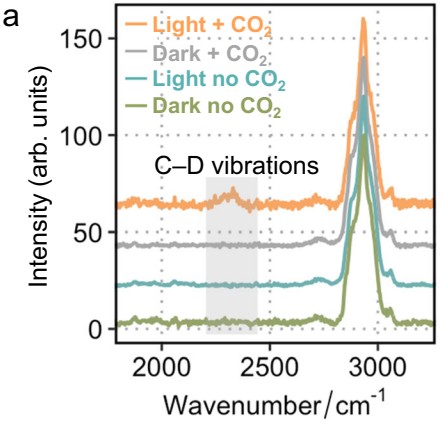
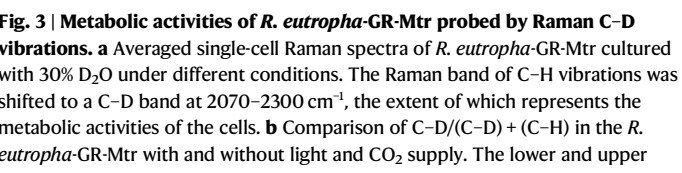
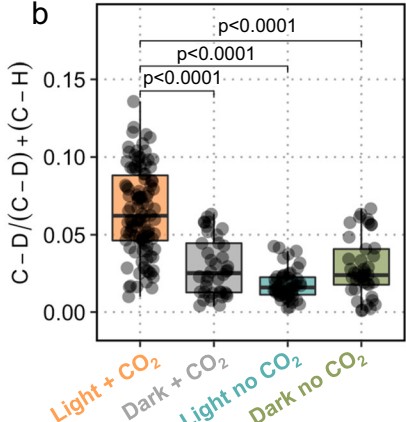

**Fig. 3 | Metabolic activities of *R. eutropha*-GR-Mtr probed by Raman C−D vibrations. a** Averaged single-cell Raman spectra of *R. eutropha*-GR-Mtr cultured with 30% $D_2O$ under different conditions. The Raman band of C−H vibrations was shifted to a C−D band at 2070−2300 $cm^{-1}$, the extent of which represents the metabolic activities of the cells. **b** Comparison of C−D/(C−D) + (C−H) in the *R. eutropha*-GR-Mtr with and without light and $CO_2$ supply. The lower and upper quartiles are drawn as lines outside the box. The rectangle in the box plots represents the second and third quartiles with a line inside representing the median. The whiskers indicated the minima and maxima. Statistics were performed with a two-sided Student's $t$-test ($p < 0.0001$, the number of measured individual cells $n = 294$). Source data are provided as a Source Data file.

## Further enhancement of artificial photosynthesis

To achieve photosynthesis with light energy as the sole energy source, we employed a solar panel to drive the electrochemical platform. Instead of using a potentiostat to control the cathode potential, we designed a simple voltage regulator to adjust the applied potential generated by the solar cells (Supplementary Fig. 6a, b). The voltage regulator was employed to stabilise the cathode potential in order to avoid cathode potential fluctuations that could harm the cells.

The biomass of *R. eutropha* was chosen as the end product of photosynthesis, as it was qualified by European Union to be used as a safe source of single-cell protein for animal feed[25]. The *phaCAB* operon has been knocked out in *R. eutropha* H16 to create *R. eutropha* Δ*pha* (RHM5) mutant for maximising carbon flux towards biomass instead of polyhydroxyalkanoates (PHA) (Fig. 4a and Supplementary Table 1). Flavin mononucleotide (FMN) was added as an electron mediator to enhance the electron transfer rate which can react with MtrC[26]. Single-cell Raman analysis showed a typical band of FMN at ~1340 cm$^{-1}$ in cells expressing Mtr (Fig. 4b and Supplementary Fig. 7), indicating that exogenous FMN could bind to MtrC as the cofactor involved in electron transport. *R. eutropha* RHM5-GR-Mtr strains (Supplementary Table 1) were inoculated in bioelectrochemical reactors with an applied voltage of ~1.7 V under light or dark over 5 days (Supplementary Fig. 6c), whilst $CO_2$ was continuously pumped in as the sole carbon source. The results showed that cells grew on $CO_2$ in the presence of light, and all controls in the dark could not grow (Fig. 4f). The $OD_{600}$ of the induced RHM5-GR-Mtr increased to 0.237 over 5-day illumination (Fig. 4f). The control group did not show biomass production, ruling out the possibility of $H_2$-mediated electron transfer to support growth. These results suggest that the engineered *R. eutropha* was able to perform photoelectron-autotrophic growth and fix $CO_2$ powered by the light-driven electron transport chain. Our previous work demonstrated that addition of exogenous flavin was effective as an electron shuttle mediating electron transfer between GR-expressing *R. eutropha* and the electrode (Supplementary Fig. 8)[10]. Although the electron-shuttling molecules can pass the outer membrane and deliver electrons to the inner-membrane-bound electron transport chain, combining the electron mediator with the outer-membrane-bound Mtr proteins created a more efficient electron transfer pathway (Supplementary Fig. 8). In this study, we compared the effects of MtrCAB on cell doubling time and system faradic efficiency. Supported by the Mtr pathway, the doubling time of *R. eutropha* RHM5-GR-Mtr almost halved, and the faradic efficiency increased to 35.4% from 23.9%, compared to *R. eutropha* RHM5-GR using flavin as an electron mediator (Fig. 4g, h). The total charge transfer of the photoelectrochemical system with *R. eutropha* RHM5-GR-Mtr increased to 105 coulombs, a 54.4% improvement compared to 68 coulombs achieved by the system with *R. eutropha* RHM5-GR. These results suggest that the expression of the outer-membrane-bound MtrCAB establishes an efficient electron transfer path connecting extracellular electrons with central carbon metabolism.

Proton motive force is the key driver that determines the kinetic rate and overall performance of the system. Therefore, to further increase the proton pumping of the photosystem, canthaxanthin was added as an antenna of GR to improve the capture of light energy (Fig. 4a). The GR-canthaxanthin complex was reported to possess a 5-fold proton-pumping capacity compared to the sole GR[27]. We resuspended the *R. eutropha* RHM5-GR strains with or without canthaxanthin in an unbuffered solution and tracked the extracellular proton concentration changes over 1 min in the light (Supplementary Method 2). The presence of canthaxanthin resulted in almost doubling ΔpH compared to the strain with GR alone (Supplementary Fig. 9). Single-cell Raman analysis revealed bands at 1005, 1155 and 1517 cm$^{-1}$ in the engineered cells when canthaxanthin was introduced, in contrast to the cells without canthaxanthin, and these bands are consistent with the Raman spectra of pure canthaxanthin chemical (Fig. 4c and

Supplementary Fig. 7). The addition of canthaxanthin changed the colour of the suspension of GR-expressing cells from pink to red (inset in Fig. 4c), which is consistent with the previous study[27]. In the presence of canthaxanthin, the doubling time of the photoelectrotrophic growth decreased by 22% to 70 h (Fig. 4g) and the efficiency increased by 21% to 42.9% (Fig. 4h) compared to the control strain with GR only. This suggests that GR-canthaxanthin complex can significantly enhance light harvesting of photosynthesis in this artificial system.

In addition to the energy generation module, the module of $CO_2$ fixation (Fig. 4a) can also be improved by overexpressing a native carbonic anhydrase *can* which catalyses the interconversion of bicarbonate and $CO_2$[28]. A portion of $CO_2$ combines with water to form bicarbonate which is then transported inwards to cells, the overexpression of *can* is thus expected to accelerate the interconversion between $HCO_3^-$ and $CO_2$ and increase the $CO_2$ utilisation[29]. When $^{12}$C-formate and $^{13}$C-bicarbonate were used as the substrates to culture the *can*-expressing RHM5, single-cell Raman analysis showed isotopic shifts of the phenylalanine band at 1003 cm$^{-1}$ to 987, 975 and 961 cm$^{-1}$, indicating incorporation of $^{13}$C from bicarbonate into biomass (Fig. 4d and Supplementary Fig. 10). We further calculated the degree of the $^{13}$C incorporation and found a higher $^{13}$C content in *can*-expressing RHM5 compared to controls (Fig. 4e). In the photoelectrochemical system, the overexpression of *can* further reduced the generation time to a minimum of 67.3 h (Fig. 4f), which is comparable to that of a native anoxygenic phototrophic bacterium *Rhodopseudomonas palustris* under electrochemical conditions (~50 h)[17]. The Faradic efficiency of the engineered strain RHM5-GR-Mtr-*can* with additional canthaxanthin can reach 45.0% (Fig. 4h), indicating that our modular engineering strategies can effectively optimise artificial photosynthesis.

## Discussion

This research demonstrates that an artificial photosynthetic system can be established by redesigning a non-phototrophic bacterium *R. eutropha*. This system can be potentially extended to transform other non-photosynthetic bacteria into photosynthetic forms. The artificial photosynthetic system was inspired by natural photosynthesis. We have developed an electrochemical system powered by light energy, to split water, mimicking the function of photosystem II. To achieve the function of photosystem I for the regeneration of NADH/NADPH, rhodopsin was incorporated to generate proton motive force, facilitating the process of electron transfer (Fig. 5). Unlike common $H_2$-mediated microbial electrosynthesis[30], we engineered a transmembrane conduit Mtr complex as an interface between an extracellular cathode and the intracellular electron transport chain. Such electron transfer efficiency, in theory, could be comparable to an $H_2$-mediated system[31]. To rule out hydrogen generation and ensure that the Mtr-mediated electron transfer was the major mechanism, a relatively low potential at ~1.7 V was applied with no evolution of hydrogen detected in the headspace (Supplementary Fig. 4c). The control condition in the absence of light showed no cell growth (Fig. 4f), consistent with the previous reports that low potential conditions would be difficult to support cell growth[30,32]. The formation of the photosynthetic electron transport chain designed in this study is fundamentally similar to natural photosynthesis, in which electrons are transferred via a series of redox enzymes accompanied by proton movement, whilst avoiding the generation of intermediates such as $H_2$[30] and formate[33]. Such interactions between electrons and protons drive the photochemical reactions.

The engineered photoelectrotrophic *R. eutropha* in this study is similar to the anoxygenic phototroph *Rhodopseudomonas palustris* which is a model bacterium for studying phototrophic extracellular electron uptake[17]. Both the genetically engineered *R. eutropha* and the native *R. palustris* are phototrophic; however, their photosystems are unable to solely drive $CO_2$ fixation due to the absence of photosystem II which is essential for water splitting and electron generation. In the

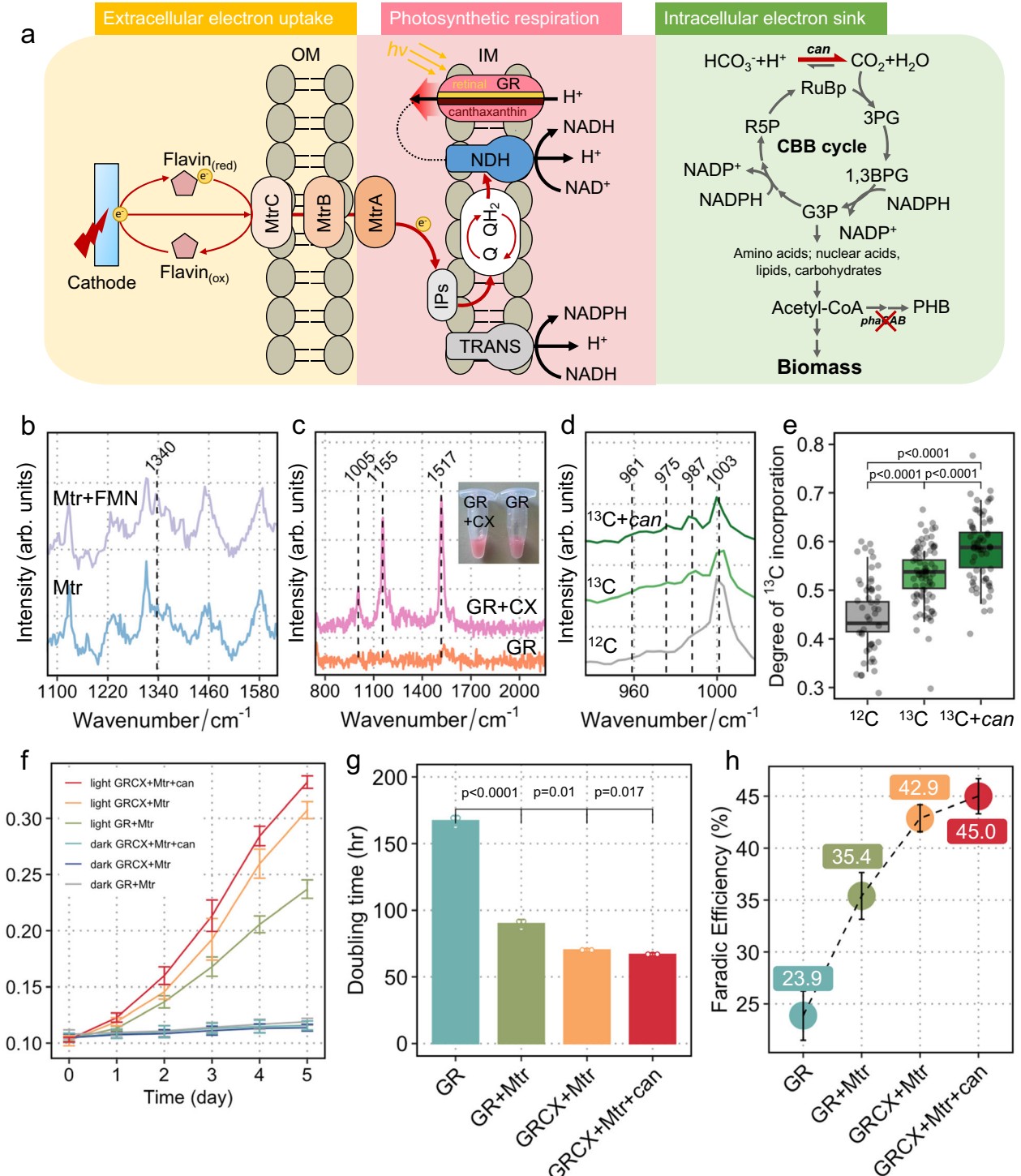

**Fig. 4 | Photoelectro-autotrophic growth of RMH5-GR-Mtr. a** The photoelectrophic system can be divided into extracellular electron uptake, photosynthetic respiration and intracellular carbon sink (OM outer membrane, IM inner membrane, IPs inner-membrane proteins, NDH NADH dehydrogenase, TRANS transhydrogenase, Q quinones). **b** A typical Raman spectrum of an RMH5-GR-Mtr cell with Mtr-bound FMN identified by a band at ~1340 cm⁻¹. **c** A typical Raman spectrum of an RMH5-GR-Mtr cell with canthaxanthin (CX) identified by three bands at ~1005, 1155 and 1517 cm⁻¹. Cell suspensions of induced RHM5-GR-Mtr with and without CX. **d** Cells grown in $^{13}C$-bicarbonate exhibited isotopic Raman shifts from 1003 cm⁻¹ to 987, 975 and 961 cm⁻¹. **e** Comparison of the degree of $^{13}C$-bicarbonate incorporation of strains with and without *can* overexpression (the number of measured individual

cells $n = 241$). The lower and upper quartiles are drawn as lines outside the box. The rectangle in the box plots represents the second and third quartiles with a line inside representing the median. The whiskers indicated the minima and maxima. **f** Growth profiles of RMH5- GR-Mtr with and without canthaxanthin under light and dark conditions (data show means ± SD of three biological replicates). **g** Comparison of generation time of strains with GR, GR-Mtr, GR-canthaxanthin complex (GRCX)-Mtr and GRCX-Mtr-*can* (data show means ± SD of three biological replicates). **h** Comparison of faradaic efficiency of strains with GR, GR-Mtr, GRCX-Mtr-CX and GRCX-Mtr-*can* (data show means ± SD of three biological replicates). Statistics were performed with a two-sided Student's *t*-test (exact *p*-values are provided in the display). Source data are provided as a Source Data file.

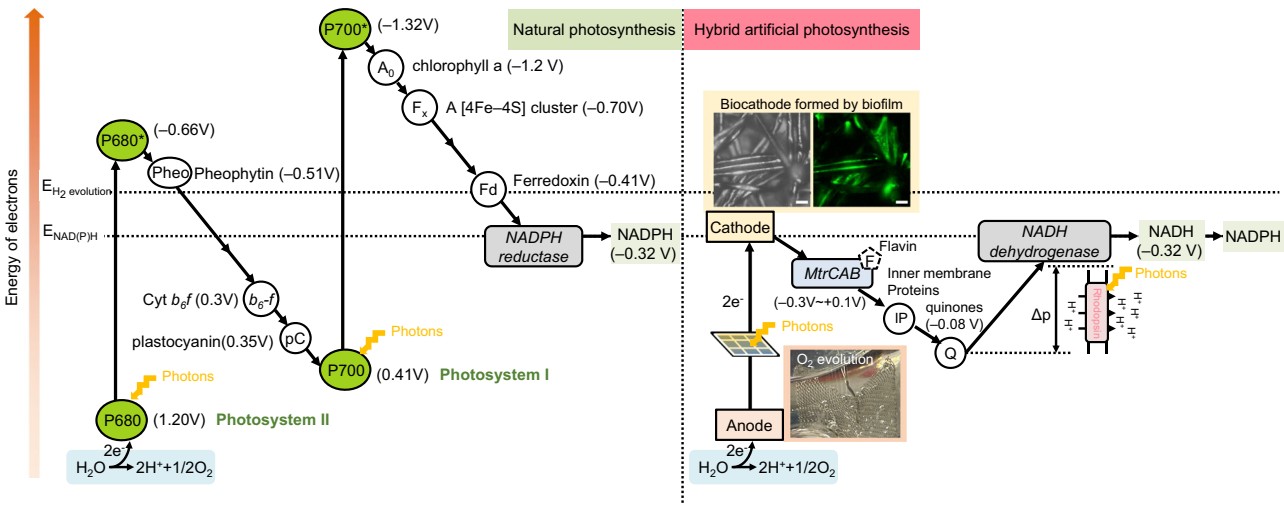

**Fig. 5 | Schematic illustration of the light reaction pathways of natural photosynthesis and designed artificial photosynthesis.** Potentials for hydrogen evolution reaction ($E_{Hydrogen\ evolution}$) and NAD(P)H ($E_{NAD(P)H}$) synthesis are highlighted by dash lines. Part of redox enzymes are omitted. Oxygen evolution with $O_2$ bubbles was observed on the anode, and biofilms were observed on the cathode with a brightfield microscopic image and FITC fluorescence image of biofilms attached to the cathode using SYTO™ 9 staining. The scale bars represent 20 µm.

presence of an electrode, both microorganisms are able to harvest electrode-supplied electrons to drive $CO_2$ fixation pathway in the light. However, the underlying mechanisms of these microbial systems are different. The engineered *R. eutropha* employs rhodopsin to generate proton motive force. We used the synthetic design to couple proton movement with electron transfer to generate the energy required for $CO_2$ fixation. In contrast, *R. palustris*'s photosystem $P_{870}$ naturally involves electron transfer and can excite electrons in the light[17]. Although the synthetic design showed comparable performance in supporting photoelectrotroautotrophic growth, further investigations are needed to deepen our understanding of the photosynthetic electron transport chain.

In the process of electron transfer in the engineered *R. eutropha*, the native biomolecule plays an important role in obtaining electrons from the Mtr pathway. In the model electroactive bacterium *S. oneidensis* MR-1, the outer-membrane MtrCAB pathway is usually coupled with an inner-membrane protein CymA to involve electron transfer[34]. However, a recent study on the engineered *E. coli* with an Mtr pathway showed that CymA might not be necessary for inward electron transport[35]. In addition, several other inner-membrane enzymes such as NapC have been suggested to perform the same function as CymA[19]. Furthermore, a recent study of *S. oneidensis* reported that hydrogenase could obtain electrons from the Mtr pathway[36]. Inspired by the finding in *S. oneidensis* MR-1, we propose that, in the native system of *R. eutropha*, the periplasm-facing membrane-bound hydrogenase could also be involved in Mtr-mediated inward electron transfer. This study demonstrates that the expression of MtrCAB complex enhances extracellular electron transfer in *R. eutropha*. Nevertheless, the enhancement in electron transfer remains limited compared to other heterologous microorganisms, such as *E. coli* which achieved an approximate 2 to 6-fold increase in the electron transfer efficiency through the Mtr pathway expression[19,37]. This observed limitation could be attributed to relatively low MtrCAB expression in *R. eutropha*. The expression of the Mtr pathway can potentially be improved by optimisation of codon and ribosome binding site, as well as the addition of precursors like 5-aminolevulinic acid[38] for heme synthesis. Additionally, the introduction of additional proteins such as periplasmic small tetraheme cytochrome (STC) may further enhance electron transfer within the periplasmic space[38]. The investigation of heterologous Mtr pathway expression in non-native bacteria has received considerable attention in recent years[19,37,38]. The introduction

of Mtr pathway has improved the extracellular electron transfer of *R. eutropha* in this study (Fig. 4h). However, the exact functionality of the Mtr expression in *R. eutropha* and other heterologous bacteria requires more rigorous verification. For example, there is a remote possibility that MtrCAB may not be directly involved in electron transfer but, instead, might increase membrane permeability, enhancing electron transfer via endogenous electron mediators such as flavins and quinones.

The reversing function of NADH dehydrogenase is key to regenerating NADH[9]. The reverse electron transfer from the quinol pool (with a midpoint potential, $E_m$, approximately −0.08 V) to NAD$^+$ ($E_m = −0.32$ V) is thermodynamically unfavourable (Fig. 5). The reversed process requires a highly reduced quinol pool to supply electrons and a high proton motive force to drive protons back through the NADH dehydrogenase in the reverse direction to forward catalysis. In this study, electrons flow inwardly from the external electrode to the Mtr pathway then leading to the reduction of the quinol pool. The effective potential of the highly reduced quinol pool should be more negative than its electrochemical midpoint potential and approached NADH potential. In the meantime, the proton motive force generated by rhodopsin facilitated the NADH dehydrogenase-mediated reverse electron flow for NADH regeneration. In *R. eutropha*, there are two types of NADH dehydrogenase, namely *nuo* and *ndh*[21]. *nuo* is a proton-translocating protein and *ndh* is a non-proton-pumping membrane-bound protein. According to a previous study of *S. oneidensis*, *nuo* is critical to inward electron transfer from the electrode for NADH generation[39]. In this system, the NADH generation was impeded by the addition of cyanide m-chlorophenyl hydrazone (CCCP), a protonophore (Supplementary Fig. 11). Therefore, it is logical to reason the significant role of the proton-translocating *nuo* in *R. eutropha* in the process of inward electron transfer for NADH generation.

In this system, the proton motive force primarily relies on the photoreaction of the proton-pumping rhodopsin. There are two strategies for enhancing the proton pumping of rhodopsin. One approach is increasing light intensities, thereby elevating the proton-pumping rate of the rhodopsin. We found that increasing light intensities led to enhanced biomass increases in *R. eutropha* RHM5-GR-Mtr (Supplementary Fig. 12). The other approach is binding an antenna molecule to rhodopsins. In nature, microorganisms cannot control the light intensities of natural light, but they synthesise the antenna molecules

to bind with rhodopsin for enhanced energy absorption and energy generation[40]. In the system presented here, the exogenous antenna molecule canthaxanthin was introduced which can be potentially biosynthesised. The addition of canthaxanthin did not adversely affect the viability of cells (Supplementary Method 3 and Supplementary Fig. 13). Furthermore, the GR-canthaxanthin complex was reported to have higher thermal stability compared to GR[27].

Reactive oxygen species (ROS) is a potential concern in photoelectrosynthesis because of their toxicity to cells[30,32,33]. In this study, a high proton motive force and reverse electron transfer could be possible factors contributing to ROS generation. To avoid the risk of ROS generation, we employed a dual-chamber electrochemical reactor separated by a Nafion proton exchange membrane (PEM) to reduce the effect of anodic oxygen generation on microbial carbon fixation in the cathodic chamber. This design ensures that oxygen produced from water splitting in the anodic chamber cannot permeate the PEM, thus preventing its entry into the cathode chamber to generate ROS. In addition, the $N_2/CO_2$ syngas was continuously pumped into the cathode chamber to maintain an anaerobic condition. Although the anaerobic cathode chamber can effectively avoid ROS generation, it could sacrifice the cellular energy generated via the aerobic respiration pathway. Therefore, a strategic approach might involve controlling oxygen levels while simultaneously enhancing microbial resistance to ROS. Achieving a balance between reducing ROS effects and improving energy generation would be important for the future scaling-up and long-term operation of the system. There are some strategies to reduce ROS when bacteria are exposed to the risk of ROS. For example, engineering bacteria to express lycopene has been shown to improve microbial resistance toward ROS[41]. The addition of antioxidants, such as glutathione, has been shown effectively to reduce ROS in the system[42].

This artificial photosynthesis system is relatively simple, in contrast to natural chlorophyll-based photosynthesis which contains complex membrane structures after millions of years of evolution. Calculations can be made with respect to the overall photosynthesis efficiency of this system. But as yet, such calculations will have to be based on so many assumptions that a meaningful number is not yet attainable. Chlorophyll does not absorb photons in the photosynthetically available radiant (PAR) waveband evenly, with only a low degree of absorbance occurring in the green region. In contrast, rhodopsin strongly absorbs green-blue light (~500 nm). The complementary light absorption offers the possibility of integrating chlorophyll and rhodopsin photosystems in either a pure culture[43] or a mixed community[4] to maximise the utilisation of solar energy. The artificial photosynthesis outlined in this research is modularised, allowing for further optimisation. In this study, the addition of flavin and canthaxanthin enhanced the electron transfer rate and the proton-pumping rate, respectively. Material engineering such as electrode modification can provide effective approaches to further increasing energy efficiency[44]. It is even possible to achieve maximum utilisation of light energy by extending rhodopsin's light absorption to non-visible wavelengths, which has already been shown feasible for the GR photosystem[45].

Many studies have demonstrated that microbial rhodopsins are major contributors to solar energy harvesting in the ocean, compared to chlorophyll[4,46–48]. Antennas such as canthaxanthin are found binding to rhodopsin and may have an important effect on rhodopsin phototrophy[40]. Rhodopsin phototrophy is recognised as independent of electron transfer and does not involve redox processes. In recent years, the proton motive force generated by rhodopsin has been demonstrated to drive NADH dehydrogenases in reverse, which couples the rhodopsin-based photosystem with the electron transport chain[9,10]. Interestingly, there is evidence showing transmission of Mtr-mediated extracellular electron transfer among oceanic bacteria[49]. In addition to the solid-phase electrode, syntrophic metabolisms[50] and

inorganic minerals[51] could be used as natural electron sources for bacteria growth if they contain both rhodopsin and Mtr complexes. In nature, the ubiquity of rhodopsins might allow microbes with Mtr pathways to use various electron sources besides just water splitting. Therefore, our demonstration of artificial photosynthesis in *R. eutropha* could guide future research into connecting rhodopsin photosystems to extracellular electron transfer systems at the gene level in oceanic environments. It is plausible to hypothesise that autotrophic microorganisms, equipped with Mtr-like proteins and rhodopsin in nature, could potentially fix $CO_2$ powered by solar energy.

## Methods

### Bacterial strains, culture conditions and plasmid construction

All bacterial strains, plasmids, and primers used in this study are shown in Supplementary Table 1 and Supplementary Fig. 14. The primers used in this study are shown in Supplementary Table 2. *E. coli* strains were grown in Lysogeny Broth (LB) at 37 °C under aeration by shaking at 200 rpm. *R. eutropha* and *S. oneidensis* strains were grown in LB at 30 °C under aeration by shaking at 150 rpm. If required, antibiotics (Sigma-Aldrich) were added as follows: 10 μg mL$^{-1}$ gentamicin and 10 μg mL$^{-1}$ tetracycline for *R. eutropha*; 12.5 μg mL$^{-1}$ tetracycline for *E. coli*. Before the bioelectrochemical experiments, *R. eutropha* strains were precultured in Tryptic Soy Broth (TSB) and a minimal medium with 20 mM fructose or 60 mM formate as the carbon source. The minimal medium was prepared and filter sterilised, composed of 6.74 g L$^{-1}$ Na$_2$HPO$_4$·7H$_2$O, 1.5 g L$^{-1}$ KH$_2$PO$_4$, 1.0 g L$^{-1}$ (NH$_4$)$_2$SO$_4$, 1 mg L$^{-1}$ CaSO$_4$·2H$_2$O, 80 mg L$^{-1}$ MgSO$_4$·7H$_2$O, 0.56 mg L$^{-1}$ NiSO$_4$·7H$_2$O, 0.4 mg L$^{-1}$ ferric citrate, 200 mg L$^{-1}$ NaHCO$_3$, 1 mL L$^{-1}$ concentrated metals solution (1.5 g L$^{-1}$ FeCl$_2$·4H$_2$O, 0.19 g L$^{-1}$ CoCl$_2$·6H$_2$O, 0.1 g L$^{-1}$ MnCl$_2$·4H$_2$O, 0.07 g L$^{-1}$ ZnCl$_2$, 0.062 g L$^{-1}$ H$_3$BO$_3$, 0.036 g L$^{-1}$ Na$_2$MoO$_4$·2H$_2$O, 0.025 g L$^{-1}$ Na$_2$WO$_4$·2H$_2$O and 0.017 g L$^{-1}$ CuCl$_2$·2H$_2$O), 10 mL L$^{-1}$ concentrated vitamin solution (2 mg L$^{-1}$ D-biotin, 2 mg L$^{-1}$ folic acid, 10 mg L$^{-1}$ pyridoxine HCl, 5 mg L$^{-1}$ thiamine HCl, 5 mg L$^{-1}$ nicotinic acid, 5 mg mL$^{-1}$ D-pantothenic acid, hexacalcium salt, 0.1 mg L$^{-1}$ cobalamin, 5 mg L$^{-1}$ p-aminobenzoic acid, 5 mg L$^{-1}$ α-lipoic acid and 5 mg L$^{-1}$ FMN) and 10 mL L$^{-1}$ concentrated amino acids solution (2 g L$^{-1}$ L-glutamic acid, 2 g L$^{-1}$ L-arginine, and 2 g L$^{-1}$ D, L-serine), and pH was adjusted to 7. For the anabolic test of the overexpression carbonic anhydrase in *R. eutropha*, the *R. eutropha* mutants with pLO11a-*can* were pre-grown overnight in the minimal medium with and without 0.1% (w/v) arabinose, respectively, using 20 mM fructose as carbon sources. Then the precultured strains were washed three times with the minimal medium and inoculated in the minimal medium with 20 mM formate and 10 mM $^{13}$C-bicarbonate (initial OD = 0.5). After 24 h of culture, cells were collected for single-cell Raman analysis to investigate the incorporation of bicarbonate.

The plasmid pLO11a-Mtr containing the MtrCAB biosynthesis gene cluster was introduced to synthesise the MtrCAB complex in *R. eutropha* H16-Mtr. The plasmid pLO11a-GR containing the *Gloeobacter* rhodopsin gene (GR) was introduced to make GR in *R. eutropha* H16-GR[10]. pLO11a-GR-Mtr containing GR and MtrCAB was introduced to *R. eutropha* Δ*pha* to make RHM5-GR-Mtr. pLO11a-GR-Mtr-*can* containing GR, MtrCAB, and *can* was introduced to make *R. eutropha* Δ*pha* (RHM5) to make RHM5-GR-Mtr-*can*. Induction of strains transformed with the pLO11a expression vector containing the arabinose-inducible P$_{BAD}$ promoter was carried out by growth to log phase and the addition of 0.1% (w/v) L-arabinose (Sigma-Aldrich) and overnight growth at 30 °C for *R. eutropha*. Induction of GR expression was accompanied by the addition of exogenous *trans*-retinal (Sigma-Aldrich) to a final concentration of 5 μg mL$^{-1}$. Exogenous *trans*-canthaxanthin (Sigma-Aldrich) dissolved in dimethyl sulfoxide was added as an antenna for GR with a final concentration of 5 μg mL$^{-1}$. Flavin mononucleotide (FMN) was added at a final concentration of 10 μmol L$^{-1}$ to investigate the interaction between outer-membrane cytochrome Mtr and FMN.

Cloning procedures were performed according to the manufacturer's instructions as follows: polymerase chain reaction (PCR) was carried out using Q5 DNA polymerase (NEB, UK) with synthesised primers (Sigma-Aldrich Co.). The construction of plasmid was performed in *E. coli* DH5α using HiFi assembly (NEB, UK). After extraction and purification using QIAprep Spin Miniprep Kit (Qiagen, UK), the plasmid was transferred into *R. eutropha* strains by conjugation.

## Single-cell Raman spectra (SCRS) measurements and analysis

Single bacterial cells, their metabolic profiles and pure standard chemicals were characterised by Raman microspectroscopy. Prior to the measurements, bacterial cells were washed three times with distilled water to remove traces of culture medium and extracellular metabolites. The cells were observed under a microscope after washing. Cells were diluted so that individual bacteria could be observed with a 1.5-μL suspension dropped onto an aluminium-coated slide and air-dried. Raman spectroscopic acquisition was performed using a LabRAM HR Evolution confocal Raman microscope using a 100×/0.75 air objective (HORIBA, UK). Single-cell Raman spectra (SCRS) were obtained using a 532-nm neodymium-yttrium aluminium garnet laser with a 300-grooves mm$^{-1}$ diffraction grating and were acquired in the range of 100–3200 cm$^{-1}$. The laser power was set at ~80 mW, which was attenuated by neutral density (ND) filters before focusing on the samples. Spectra were recorded with LABSPEC 6 software (HORIBA, UK). All raw spectra were pre-processed by cosmic ray correction, polyline baseline fitting and subtraction and vector normalisation of the entire spectral region. Linear discriminant analysis (LDA) was used for dimension reduction of single-cell Raman spectroscopy to aid visualisation at the single-cell level. All analysis and plotting were done under an R 4.2.2 environment.

## Analysis and quantification of intracellular biomolecules

Single-cell Raman spectroscopy was employed to characterise biomolecules of cytochromes, rhodopsin, and canthaxanthin-GR as well as C–D band and phenylalanine band for stable isotope probing. A green incident laser light with a wavelength of 532 nm was employed for our Raman measurements. Importantly, all the biomolecules that we characterised with Raman spectroscopy absorb visible light at wavelengths very close to the incident light. For example, cytochromes absorb light at 530 nm[52], rhodopsin at 520 nm[53], and canthaxanthin-GR at 485 nm[27]. This proximity in wavelengths results in a resonance effect, which significantly enhances the Raman scattering cross-section and consequently improves the signal-to-noise ratios by orders of magnitude. Furthermore, the phenylalanine band selected for $^{13}$C-labelling experiments contains an aromatic ring that also contributes to this resonance effect. The deployment of Raman-deuterated water has been widely applied in probing microbial activities[54] and offers a unique, universally applicable approach for non-destructively measuring single-cell activities[23].

For characterising cells under induced or uninduced conditions with arabinose, triplicates were performed in each condition and Raman spectra were acquired using a 25% power filter and 3 to 5-s acquisition time with high signal-to-noise ratios. Raman bands at 748 (pyrrole breathing mode), 1128 (υ(CN) stretching vibrations), 1312 (δ (CH) deformations) and 1584 cm$^{-1}$ (υ (CC) skeletal stretches) were attributed to cytochromes. Quantification of cytochrome *c* was achieved by integrating band areas at each of these four band positions. Intracellular phenylalanine was identified and quantified by the phenylalanine band centred at 1003 cm$^{-1}$. Raman spectra of pure riboflavin 5′-monophosphate sodium salt hydrate (Sigma-Aldrich, UK) in powder as well as in water solution was measured as a standard to identify a band at 1340 cm$^{-1}$ as an indicator for quantification of flavin, which is assigned to the middle ring vibration of the tricyclic isoalloxazine structure of the flavin. *Gloeobacter* rhodopsin (GR) complexes were characterised by Raman microscopy using a 1% power

filter and 1-second acquisition time. The low power and short acquisition time were used to prevent photobleaching of the GR chromophores. SCRS of strains with the pLO11a expression vector, either with or without the addition of L-arabinose were measured in triplicates. GR complexes were identified by a band at ~1530 cm$^{-1}$ above the background noise in the SCRS. Quantification of biomolecules was done by integrating the area of the corresponding Raman bands and represented as box plots. The lower and upper quartiles are drawn as lines outside the box. The rectangle in the box plots represents the second and third quartiles with a line inside representing the median. Sample means were compared by using Welch's two-sample *t*-test for unequal variance.

The degree of $^{13}$C incorporation into biomass[24] was determined by calculating the isotopic shifts of phenylalanine from 1003 cm$^{-1}$ to 987, 975, and 961 cm$^{-1}$. Depending on different $^{13}$C substitutions on the phenylalanine ring, a total of four possible Raman positions of the $^{13}$C/$^{12}$C mixture existed for possible isotopomers. Due to the symmetric structure, substitutions at only three carbon sites on the phenyl ring can affect the vibrational modes hence the wavenumbers: the bands at 1003 cm$^{-1}$ correspond to structures where all three carbon are $^{12}$C; the bands at 987 cm$^{-1}$ occur when any of the three carbon sites are substituted with $^{13}$C; the 975-cm$^{-1}$ bands occur when any two of the sites are $^{13}$C; and the bands at 961 cm$^{-1}$ only appear when all three carbon sites are $^{13}$C and hence occur only with high assimilation from $^{13}$C-bicarbonate into biomass. The degree of $^{13}$C incorporation was calculated by the ratio of total $^{13}$C to total $^{13}$C and $^{12}$C as follows, in which A represents the area under the curve centred at a defined wavenumber:

$$\text{Total}\,^{13}\text{C incorporation} = \frac{A_{961cm^{-1}} + \frac{2}{3}A_{975cm^{-1}} + \frac{1}{3}A_{987cm^{-1}}}{A_{975cm^{-1}} + A_{987cm^{-1}} + A_{1003cm^{-1}}} \quad (1)$$

## Determination of NADH and NADPH levels

For measurements of intracellular reductant levels including NADH and NADPH, cells were harvested from the working chamber and pre-treated according to the manufacturer's instructions. NADH/NAD$^+$ and NADPH/NADP$^+$ ratios were determined using an NAD/NADH Assay Kit (Abcam, USA) and an NADP/NADPH Assay kit (Sigma-Aldrich, UK), respectively. They were measured using a colourimetric assay with a microplate reader (BioTek Corporation, UK) equipped with Tecan SPARKCONTROL Dashboard software to detect the absorbance at a wavelength of 450 nm.

## Fluorescence microscopy to visualise biofilm on the electrode

The Raman microscope chassis was modified with an epifluorescence system in consultation with the manufacturer (HORIBA, UK). An LED lamp and a FITC filter block were used to visualise the fluorescence of the engineered cells stained by SYTO™ 9 (Thermo Fisher Scientific, UK) with a 20×/0.4 objective. The final images were processed with background denoising using ImageJ2 (version number: 2.3.0/1.53 f).

## Microbial photoelectrochemical system

A dual-chamber bioreactor (70 mL of total volume) separated by a Nafion membrane (only allowing proton transfer) was used as a bio-photoelectrochemical system. The microbial photoelectrochemical experiments were performed in a three-electrode configuration on a multichannel potentiostat (PalmSens, Netherlands). For the anode chamber, a stainless-steel mesh was used as the counter electrode. For the cathode chamber, the working electrode was made of 2.5 × 2.5 cm$^2$ carbon paper (H23, Quintech, UK) and an Ag/AgCl reference electrode (3 M KCl, RE-5B, BASi, USA) was installed for measuring the potentials. The anodic electrolyte was made of 50 mM KH$_2$PO$_4$ and 50 mM K$_2$HPO$_4$ (pH = 7.2), and the cathodic electrolyte was the same as the above-mentioned minimal medium. The working chamber was

continuously agitated by a magnetic stirrer at 30 °C. For the photo-electrochemical experiments, 5 m of white LED light strips encircled the exterior of the working chamber to ensure even light distribution during the experiment, and illumination intensity was monitored by a photometer.

### Electrochemically driven nitrate reduction

To test electron transfer from the electrode to cells, arabinose-induced and uninduced *R. eutropha*-Mtr were inoculated to the cathode of the photoelectrochemical system. *R. eutropha*-Mtr strains were first activated 24 h in TSB in the presence of 10 μg mL$^{-1}$ tetracycline. Then 200 μL of bacterial cultures were inoculated into 10 mL of the fresh minimal medium on 20 mM fructose with 0.5 g L$^{-1}$ NaNO$_3$ and 10 μg mL$^{-1}$ tetracycline. After 16-hour culture, the carbon source was switched to 60 mM formate for overnight incubation. For the Mtr induction group, 0.1% (w/v) L-arabinose was added to induce the gene expression. After overnight preculture in a shaker with 150 rpm at 30 °C, the cells were harvested by centrifugation at 3000 × g for 3 min and washed three times with the minimal medium to remove organics. The cell pellets were resuspended in the fresh minimal medium and the OD$_{600}$ was adjusted to ~0.5 before transfer to the working chamber which was continuously bubbled with N$_2$ gas to maintain an anaerobic environment. Cells were subjected to anodic conditions at +200 mV$_{Ag/AgCl}$ for 2 days to exhaust any potential internal electron storage, such as PHB, and increase cell adhesion to the electrode. After acclimation, the electrode potential was switched to the cathodic condition at −500 mV$_{Ag/AgCl}$. At an intermediate time point, 20 mM of sodium nitrate was added to monitor the current change by a potentiostat (PalmSens, Netherlands).

### Electrochemically driven photosynthetic electron transport chain for reductant generation

*R. eutropha*-GR-Mtr was used to test whether the electrode-supplied electrons could be used to regenerate reducing power NADH and NADPH. The uninduced and induced strains were precultured as above and inoculated into the microbial photoelectrochemical system with an initial OD of ~0.5. During the experiments, the working chambers were poised at −500 mV$_{Ag/AgCl}$ and bubbled with N$_2$. The reactors were illuminated with a white LED light (~150 μmol s m$^{-2}$) or covered with aluminium foil as dark control. After 2 days of *R. eutropha*-GR-Mtr incubation in the light and dark, cells were harvested from the working chamber for the measurement of NADH and NADPH. To investigate the effect of the proton motive force on the reversing function of NADH dehydrogenase, the induced *R. eutropha*-GR-Mtr strains were incubated in the photoelectrochemical system with illumination for 24 h and then 100 μM of the proton motive force inhibitor, carbonyl cyanide m-chlorophenylhydrazone (CCCP) was injected to the working chamber for additional 24-hour incubation. NADH/NAD$^+$ ratios were compared between the CCCP-treated group and the control group with the blank solvent dimethyl sulfoxide.

### The use of heavy water (D$_2$O) to probe phototrophic metabolisms

The *R. eutropha*-GR-Mtr was precultured as described above. The 30% D$_2$O (v/v) minimal medium was prepared with 99% D$_2$O (v/v) (Sigma, UK) to replace the spent minimal medium in the working chamber and resuspend the cells pellet. The initial OD was adjusted to ~0.2 before being inoculated into the working chamber which was sparged with N$_2$ to remove oxygen and then changed to CO$_2$. After 2 days of incubation, single-cell Raman analysis was used to detect the C−D vibration in the cells.

### A hybrid system for light-driven CO$_2$ fixation into biomass

For light-driven experiments, a two-electrode configuration was performed with the carbon paper as the cathode and platinum as the anode. A commercial polycrystalline solar cell (1.5 W, 140 × 180 mm, RS, UK) was combined with a laboratory-assembled voltage regulator to power the water splitting to generate electrons. To maximise the carbon conversion to biomass, the RHM5 strain, an *R. eutropha* mutant without PHB biosynthetic pathways was used for biomass growth. The precultured RHM5-GR-Mtr strains were treated as above before being inoculated into the cathode chamber. Then cells were pre-grown in the cathode chamber under formatotrophic conditions with 60 mM formate to form a biofilm on the electrode. After 2 days of incubation when the biocathode was established, the spent medium was replaced with fresh medium, and the biomass (OD$_{600}$) of the planktonic cells was adjusted to ~0.1. The growth experiments were conducted by continuously sparging with an 80:20 mixture of N$_2$:CO$_2$ gas under light and dark conditions, with induction of GR, GR-Mtr or GR-Mtr-*can*, respectively. Flavin mononucleotide was introduced as an electron mediator to boost electron transfer. ~10 μmol L$^{-1}$ of flavin was added at an interval of 24 h to reduce flavin's photochemistry.

### Calculations

The electron transfer efficiency ($\eta_{electron}$) of the biomass formation could be calculated by the Eq. 2,

$$\eta_{electron} = \frac{nzF}{\text{Charge passed (C)}} \quad (2)$$

where *n* is the amount of biomass (using the formula CH$_{1.77}$O$_{0.49}$N$_{0.24}$[30] and 1 OD$_{600}$ corresponding to 0.448 g L$^{-1}$ dry biomass) product (mol), *z* is the number of transferred electrons (*z* = 4 for CO$_2$ conversion to biomass), and *F* is the Faraday constant (96,485 C mol$^{-1}$).

The doubling time (g) of the strains could be calculated by the Eq. 3,

$$g = \frac{\ln 2}{\ln(OD_{day5}/OD_{day1})/4} \quad (3)$$

where OD$_{day5}$ and OD$_{day1}$ represent the OD values of the planktonic cells on Day 5 and Day 1, respectively.

### Reporting summary

Further information on research design is available in the Nature Portfolio Reporting Summary linked to this article.

## Data availability

Data supporting the findings of this work are available within the paper and its Supplementary Information files. A reporting summary for this Article is available as a Supplementary Information file. Source data are provided in this paper.

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

## Acknowledgements
W.E.H. thanks EPSRC (EP/M002403/1 and EP/N009746/1) for financial support. W.E.H. gratefully acknowledges funding from EPSRC (EP/M02833X/1) for instrumentation. We thank Prof Kwang-Hwan Jung at Sogang University for providing plasmids of beta-carotene and GR and had helpful discussions. We thank Harris Saeed for making the voltage regulator. We also thank Oliver Lenz at Technische Universität Berlin, Germany for providing wild-type *Ralstonia eutropha* H16 and cloning plasmids.

## Author contributions
W.E.H. conceived the original idea; W.T. and W.E.H. designed research; W.T. performed research; I.P.T. contributed reagents and analytic tools; W.T., J.X. and W.E.H. analysed data; W.T. and W.E.H. drafted the manuscript, and all authors revised the manuscript.

## Competing interests
W.T., I.P.T. and W.E.H. have filed a provisional patent application, through the University of Oxford, with the UK Patent Office related to this work. The patent application title is "Engineering of a Photoautotrophic Cell for $CO_2$ Fixation", and the International Patent Application Number is PCT/GB2023/052452. The specific aspects of the manuscript encompassed by the patent application include methods for engineering rhodopsin-based photoautotrophic $CO_2$ fixation. J.X. declares no competing interests.
