## [Peer Review File · Nature Communications]

Engineering artificial photosynthesis in *Ralstonia eutropha* for CO₂ fixationReviewers' Comments:

Reviewer #1:

Remarks to the Author:

In this study the authors revisit their work on creating an artificial system for photosynthesis, here composed of an external electronic circuit containing a PV cell, in combination with chemo/litho-autotrophic bacteria expressing a bacterial rhodopsin. In this approach the authors are building upon the work described in their previous publications, e.g. as cited in ref 10.

The paper is reasonably well-written, although it would clearly benefit from English style editing. It describes in good detail a set of experiments that have been carried out with some very powerful experimental techniques, like single-cell Raman spectroscopy, thus leading the authors to claim that: "The artificial photosynthesis developed in this study could help design alternative methods for CO₂ fixation and produce valuable chemicals. The photo-electrotrophic system could contribute to the development of new sustainable strategies, ultimately aiding in the reduction of global CO₂ emissions". However, it is my opinion that this claim is not warranted by the data presented and is far too optimistic.

Nevertheless, the authors are presenting four aspects in this work that go beyond that what was presented in ref 10: They: (i) Equip the GR containing cells with an Mtr-based electron transfer system for passage of electrons over the cell envelope; (ii) added the carotenoid canthaxanthin as an antenna for GR; (iii) expressed a carbonic anhydrase in the artificial photo-autotroph for better CO₂ fixation; and (iv) added a PV cell to the system. None of these elements is really new in this field, but jointly they have not been put to use before.

My estimate is, however, that there are major uncertainties/mistakes in the interpretation of the results obtained, as described in the current manuscript. The first is that the authors use sunlight twice with maximal efficiency for their calculations (while we only have one sun). The second is that the demonstrated functionality of the Mtr system in nitrate respiration cannot automatically be accepted as proof for its functionality in the GR-driven photosynthesis; in the latter system the macroscopic electrode is held at such a negative potential (relative to a silver-based reference electrode), that is could even generate the NADPH/NADP⁺ ratio to a potential of ~ -435 mV. There is a (remote) possibility that engineering the Mtr system is accompanied by an increased outer membrane permeability for the external redox mediators (i.e. FMN and riboflavin), that could be at the basis of the increased rates of electron transfer. Furthermore, the chain of electron/hydrogen carriers that has been created implies the involvement of non-physiologically large redox gaps, so that long-term functioning would be seriously endangered by damage due to ROS formation.

A few additional minor points of criticism:

Line ... : we also have the bilins and the carotenoids for this

Line 239: The can enzyme is not driving the direction of the reaction; it is only catalyzing it

Line 248: Many phototrophic bacteria have doubling times below 10 or even 5 hrs; the example cited I far from typical.

Line 258: "therefore contributing to sustainable biomanufacturing": overly optimistic as long as an MCP of the product is not provided.

Line 282: "in contrast to the efficiency of natural photosynthesis ($\sim 1\%$)" – why are the authors so pessimistic about natural photosynthesis?.

Line 288: "consortia for maximizing the light-harvesting spectrum, increasing the overall light energy transfer efficiency" – it is important to realize that although chlorophyll-based photosynthesis shows increased absorbance in certain wavelength regions, there is no part of the PAR region where there is no absorption. Hence it is straightforward to grow these organisms so that they absorb all available PAR light.

Line 307: "Therefore, our demonstration of artificial photosynthesis in *R. eutropha* could unveil the presence of extensive yet undocumented rhodopsin phototrophs in oceanic environments" – for me this is too much speculation; finding such organisms via gene probes to me seems like a much more rational approach.

Line 353: "Flavin mononucleotide (FMN) was also added to be anchored on the Mtr with a

concentration of 10 $\mu\text{mol/L}$ " – please explain what 'anchored' means here.

Line 437: Please explain: "5 m of white LED light strips were set adjacent to the working chamber".

Line 453: "to exhaust any potential internal electron storage, such as PHB" – it is not sufficiently clear where the mutant strain was used that could not synthesize PHB.

Line 491: "10 $\mu\text{mol/L}$ of flavin was added at an interval of 24 h to reduce their photochemistry". Please explain what "their" refers to in this sentence.

Reviewer #2:

Remarks to the Author:

Summary:

In this manuscript, Tu et al. describe their engineering of *Ralstonia eutropha* for growth on CO_2 and a cathode, using light as a supplemental energy source. *R. eutropha* is capable of carbon fixation but cannot naturally interact with electrodes or use light energy, so a transmembrane electron conduit (Mtr) from *Shewanella oneidensis* and a rhodopsin from *Gloeobacter* (GR) were heterologously expressed. The manuscript presents evidence for functional expression of both Mtr and GR. The engineered strain was capable of growth using only light, the cathode, and CO_2 , demonstrating a significant advance for microbial electrosynthesis. Further strain improvements were also made, including adding a carbonic anhydrase and removing the pathway for PHB production.

Major comments:

1. Can the authors speculate on the identity of the inner membrane proteins that transfer electrons from MtrA to the quinone pool in *R. eutropha*? It would also be useful to note the types of NADH dehydrogenases present in *R. eutropha* and speculate on which is/are involved in inward electron transfer.

2. The engineered metabolism appears pretty similar to what has been reported for *Rhodospirillum rubrum*. While one study on this organism was cited in the discussion, I think it would be helpful to also mention *R. rubrum* in the introduction and describe the similarities and differences between it and the engineered system.

3. The representations of the transhydrogenase and the NADH dehydrogenase in figures 3 and 4 seem somewhat misleading. Since both of these couple electron transfer and proton transfer, I think they should be represented as spanning the entire membrane.

Minor comments:

The references should be checked for correct italicization and capitalization.

The 'c' in cytochrome c should be italicized throughout.

Line 19: change 'out' to 'outer'

Line 72: change 'heterogeneously' to 'heterologously'

Line 103: I think this should be 'inset' instead of 'inlet'. There are also other instances

Line 323: remove 'as'

Line 432: indicate supplier and catalog number for the carbon cloth

Line 434/435: indicate initial pH

Supplementary line 39: change 'streaains' to 'strains'

Reviewer #3:

Remarks to the Author:

The authors engineer an artificial pathway for light-driven CO_2 fixation in *R. eutropha*. Heterologously expressed rhodopsin is used to create a proton gradient that, when combined with electron flow from heterologously expressed MtrABC, reverses the NADH dehydrogenase activity. The generated NADH is

converted to NADPH to promote CO₂ fixation via the Calvin cycle. The authors further enhance light-harvesting and electron transfer through canthaxanthin and flavin addition, respectively. Enhanced carbonate to CO₂ conversion was also achieved through carbonic anhydrase expression. The authors rely on Raman measurements to confirm protein expression and enhanced metabolic activity through C13 and deuterium-labeling.

Previous studies have enhanced NADH production through rhodopsin expression in *R. eutropha* (using a mediator for electron transfer for carbon fixation applications, Huang et al.) and in *S. oneidensis* (using its native MtrCAB pathway for electron transfer, TerAvest et al.). The novelty in this work is the co-expression of the MtrCAB pathway in the rhodopsin-engineered *R. eutropha*. To my knowledge, the MtrCAB pathway has not been previously expressed in *R. eutropha*, and the application of the MtrCAB pathway for CO₂ fixation in any host has yet to be demonstrated. This study thus opens the doors to an impactful and previously unexplored application of this pathway which I believe aligns well with the high standards of this journal. However, the Raman analysis does not provide a convincing basis for supporting the conclusions of this study, and the authors are missing comparisons between their new strain and the previous strain without the MtrCAB pathway. I therefore recommend publishing this manuscript with major revisions.

Major:

1. SDS-PAGE is needed to confirm correct protein size and localization of heterologous proteins, as well as heme insertion for the MtrCAB complex.
2. Raman analysis has a high signal to noise ratio. The authors should corroborate their measurements (including protein expression and isotope labeling) with more sensitive techniques.
3. The MtrCAB complex is not significantly effective in promoting charge transfer in the absence of inner membrane (CymA) or periplasmic (STC) expression (see Jensen et al. DOI: 10.1021/acssynbio.5b00279; Mouhib et al. DOI: 10.1101/2021.08.28.458029). It is unclear if this strain can outperform the previous rhodopsin strain without the MtrCAB expression, especially since this strain does not have a pathway for transfer from the outer membrane. The performance of this strain needs to be compared to the previous rhodopsin strain without MtrCAB (with and without a mediator).

Minor:

4. The text should be reviewed to make sentences more concise and easier to read. Some examples:
 - "Microbial rhodopsins as the most widespread phototrophic system throughout the microbial world are hypothesised to have a significant role in microbial carbon fixation which is beyond our current knowledge" to "Microbial rhodopsins, the most widespread phototrophic proteins in the microbial world, are hypothesised to have a significant role in microbial carbon fixation."
 - "...rhodopsin phototrophy was proven to generate..." – replace "proven" with "shown"
 - "...rhodopsin cannot solely drive the CO₂ fixation pathway by its proton transport." – replace "by" with "with"
 - "Such Yin (electrons) and Yang (protons) interactions drive the photochemical reactions in this artificial photosynthesis." – analogy to Yin and Yang interactions are not necessary
5. Relabel "Discussion" to "Conclusion"
6. Correction of typos and improvement in figures. Some examples:
 - Fig. 1 change "Nafin" to "Nafion", and label anode and cathode compartments in figure
 - Fig. 2 – abbreviations (IN, OM, IP) should be explained in caption, spacing before "(d)" needs to be added in caption, caption of (f) "percentage of cells" is unclear (is this the percentage in the number of cells in each culture that is expressing vs. those that aren't? how is this determined?)
 - Fig. 4 – GRCX needs to be specified

Reviewer #4:

Remarks to the Author:

Summary & Noteworthy results. This work builds on a previously established strain of *Ralstonia eutropha* (*Cupriavidus necator*) H16 engineered to express rhodopsin from *Gloeobacter* (reference #10, Davison PA, et al). The primary innovation was the introduction of an extracellular electron transfer conduit known as the multi-heme transmembrane reductase (MtrCAB), which enabled faster extracellular electron transfer into *R. eutropha*'s native internal electron transfer chain (ETC) via a redox mediator (riboflavin) and external electrode. The authors claim the flow of low potential electrons into the quinone pool via the MtrCAB, coupled with the inverted proton motive force driven by the light-activated rhodopsin, drives the native ETC in reverse. The authors suggest this ultimately drives more NADH and NADPH synthesis, in a physiologically reversed manner, which is then utilized in the CBB cycle to boost CO₂ capture and biomass production. The work uses rigorous electrochemical and Raman, UV-VIS spectroscopic methods to monitor Mtr functional expression, metabolic activity, and cell growth. The method and the results are well-presented and interesting. However, I find the interpretation of the results far-fetched, and unsatisfactorily unsupported by the literature. Significance to the field, comparison to established literature & Originality. The host system and its characterization with an electrode is partially original, as it builds upon a previously existing system that uses a transmembrane-mediated electron transfer that proceeds via an unclear mechanism (ref 10). Secondly, while the incorporation of extracellular ET pathways for electrode-driven biosynthesis has also been shown (ref 9: Tefft NM, TerAvest MA.), this article presents the first expression of both proteins in this particular host organism, *Ralstonia eutropha*. The functional incorporation of MtrCAB into this host is indeed an innovative step forward and will be significant for future systems incorporating such an extracellular ET pathway.

Does the work support the conclusions and claims, or is additional evidence needed? Are there any flaws in the data analysis, interpretation and conclusions? The methodology is rigorous, the presentation of the data is high quality and meets the standards of the journal, and there is enough detail provided in the methods for the work to be reproduced. Overall, the authors apply a rigorous scientific method and present their results with great clarity. However, the interpretation of the results, particularly related to the mechanistic hypotheses, are seemingly not feasible and far-fetched. The mechanism of 'artificial photosynthesis' via a pmf-driven voltage-up conversion in NAD(P)H dehydrogenase is, according to my knowledge, unprecedented. While it surely is an interesting hypothesis, I cannot rationalize a physiological basis for this seemingly complex and inverted reaction to exist, nor can I find any supporting literature outside of the authors' own previous claims. This issue prohibits publication and requires revision. I will elaborate on this issue in the text that follows. Major issue: The interpretation of increased biomass production via voltage upconversion via a pmf.

How does the quinone inject an electron into the NAD(P)H dehydrogenase? It is a ~250 mV thermodynamically unfavorable jump, according to Figure 5. The authors have shown the midpoint potential of the Quinone (-0.08V). However, I would assume the effective Quinone/Quinol potential to be much more negative, as the quinone pool is largely reduced by the MtrCAB supplied low potential electron flux. Perhaps that could explain why ET is injected into NAD(P)H dehydrogenase.

How does a proton gradient drive NADH reduction? I cannot rationalize how the light-activated GR transfer of protons into the lumen will drive NAD⁺ to NADH reduction (which is physiologically reversed). In fact, removing the protons from that side of the membrane should make NAD⁺ + H⁺ + e⁻ → NADH even more difficult, as protons are being depleted from this reaction. The authors must rationalize this mechanism or provide an alternative hypothesis before I can accept this manuscript. For instance, can the authors exclude the hypothesis that the light-driven GR-proton pump is simply driving a pmf that drives more ATP synthesis, and the additional electrons from the MtrCAB are helping drive electrons into NAD(P)H dehydrogenases?

Line 136: The authors cite that previous studies that they claim 'prove' pmf driven voltage up conversion. First and foremost, in science one can never definitively 'prove' anything, you can offer the best and most well-informed explanation. Secondly, are there indications from other literature sources that support this pmf-upconversion hypothesis? Specifically, is there any evidence to suggest that NAD(P)H dehydrogenase activity is dependent upon a pmf? E.g. Are the midpoint potentials of the cofactor in NADP dehydrogenase pH dependent? For ATP synthase this is clear, the pmf and flux of

protons results in conformation changes of ATP synthase that drive ATP synthesis. However, I can find no such conformational change in NAD(P)H dehydrogenase, and no support from the literature that NADH activity is pmf dependent. Thirdly, the fact that your previous mechanistic assumptions are now published doesn't mean they are now definitive 'proof'.

Minor issues. Can the authors please specify why expression of this MtrCAB/Rhodopsin in *R. eutropha* was novel or even advantageous when compared to the previous study in *Shewanella oneidensis* (reference #9)

Line 86; can the authors provide quantitative values for the improvements of ET rates and proton pumping capacity? If not, can the authors quantitatively reveal this in terms of improved growth, not just growth rate, but overall yield?

Line 234: The authors state 'canthaxanthin' can 'significantly enhance' photosynthesis. But I would argue that the difference, while statistically 'significant', is very minor (just a few %). Please write the quantitative % increase instead.

Line 247: a ~2.7 hour increase in doubling time (from 70h without to 67.3h with the addition of carbonic anhydrase) is minor in my opinion. Furthermore, batch to batch variability is not reported, what is the n-value here? I do not see it in the Figure 4 caption, in which it should be added.

Line 56- poor grammar, the sentence is difficult to understand.

Abstract: I find it inconsistent that the authors list the species name for the host strain, the rhodopsin (that was previously incorporated), but omit the species name of the newly introduced MtrCAB electron transfer conduit in the abstract (which is their primary novelty in this work).

Lack of clarity: From the abstract alone, it was not easy for me to pinpoint what innovation the authors made, without diving into their previous work. Only upon reading the following in the intro: 'In this study, *R. eutropha* H16 was engineered to heterogeneously express the MtrCAB 73 complex and GR protein' could I begin to identify their novelty. Due to the extensive engineering of this strain, the authors should clarify what was incorporated in the strain previously (perhaps already in the abstract), and what specific novelty was added in this work (the MtrCAB).

General comments: The authors increased GR's light-harvesting capacity using a dye (canthaxanthin). I am curious if the authors have thought of doing a simple power study, in which growth under increased (or decreased) LED intensity was monitored. A conclusion on where light harvesting becomes rate limiting could be reached using such a method. Off-target effects of the dye, e.g. local heating, are not discussed.

REVIEWER COMMENTS

Reviewer #1 (Remarks to the Author):

In this study the authors revisit their work on creating an artificial system for photosynthesis, here composed of an external electronic circuit containing a PV cell, in combination with chemo/litho-autotrophic bacteria expressing a bacterial rhodopsin. In this approach the authors are building upon the work described in their previous publications, e.g., as cited in ref 10.

1. The paper is reasonably well-written, although it would clearly benefit from English style editing. It describes in good detail a set of experiments that have been carried out with some very powerful experimental techniques, like single-cell Raman spectroscopy, thus leading the authors to claim that: “The artificial photosynthesis developed in this study could help design alternative methods for CO₂ fixation and produce valuable chemicals. The photo-electrotrophic system could contribute to the development of new sustainable strategies, ultimately aiding in the reduction of global CO₂ emissions”. However, it is my opinion that this claim is not warranted by the data presented and is far too optimistic.

Response: We thank the reviewer's valuable feedback. In response to the comment, we have revised the claim to provide a more accurate representation of our study's findings. We have revised the claim to make it more specific and accurate : “Overall, the artificial photosynthesis developed in this study could help design alternative methods for CO₂ fixation, providing a valuable reference for engineering non-photoelectrosynthetic bacteria into photoelectrotrophs, and offer insights into future investigations of the potential interplay between rhodopsin-based metabolism and extracellular electron transfer, along with their joint effect on microbial CO₂ fixation.”

Nevertheless, the authors are presenting four aspects in this work that go beyond that what was presented in ref 10: They: (i) Equip the GR containing cells with an Mtr-based electron transfer system for passage of electrons over the cell envelope; (ii) added the carotenoid canthaxanthin as an antenna for GR; (iii) expressed a carbonic anhydrase in the artificial photo-autotroph for better CO₂ fixation; and (iv) added a PV cell to the system. None of these elements is really new in this field, but jointly they have not been put to use before.

Response: We appreciate the reviewer's thorough assessment and the concise summary of the key advancements in this work compared to our previous research.

2. My estimate is, however, that there are major uncertainties/mistakes in the interpretation of the results obtained, as described in the current manuscript. The first is that the authors use sunlight twice with maximal efficiency for their calculations (while we only have one sun).

Response: In our study, we aim to evaluate the efficiency of converting solar energy into biomass in the microbial photo-electrotrophic system. Our approach involves utilising sunlight in two different ways to harness its energy, rather than counting it twice. We have considered the energy absorbed by rhodopsin, which contributes to proton motive force generation, and the energy absorbed by a solar panel for electricity generation. It's important to note that this energy originates from a single light source. To clarify this approach, we have emphasised in the Materials and Methods that our calculations are based on energy output divided by energy absorbed, as follows: “The solar-to-biomass efficiency ($\eta_{\text{solar-to-biomass}}$) of the system

could be calculated by dividing the energy output by the energy absorbed, as shown in the following equation...”.

3. The second is that the demonstrated functionality of the Mtr system in nitrate respiration cannot automatically be accepted as proof for its functionality in the GR-driven photosynthesis; in the latter system the macroscopic electrode is held at such a negative potential (relative to a silver-based reference electrode), that is could even generate the NADPH/NADP⁺ ratio to a potential of ~ - 435 mV. There is a (remote) possibility that engineering the Mtr system is accompanied by an increased outer membrane permeability for the external redox mediators (i.e., FMN and riboflavin), that could be at the basis of the increased rates of electron transfer.

Response: We understand the reviewer’s concern. The nitrate respiration experiment is to demonstrate the Mtr-mediated electron transfer from the external electrode to nitrate reductase which is a portion of the electron transfer chain of *R. eutropha*. We carried out experiments using D₂O to probe *R. eutropha*-GR-Mtr metabolism indicated the presence of Mtr, GR, and CO₂ enable the GR-driven photosynthesis, as evidenced by a C-D band observed through single cell Raman spectroscopy. Notably, no mediators (e.g., flavin) were introduced in these experiments, thereby establishing Mtr-mediated electron transfer as the predominant pathway. We acknowledge that the functionality of the Mtr system may not have been adequately demonstrated due to the absence of a control group without Mtr. To address this, we conducted additional control experiments using D₂O to investigate electron transfer in *R. eutropha*-GR strains lacking the Mtr pathway. The results showed that under both light and dark conditions, no significant C-D bands were observed in the *R. eutropha*-GR strain without Mtr, supporting the point that Mtr plays a crucial role in GR-driven photosynthesis. The result has been added into the revised manuscript as supplementary fig.5.

Supplementary fig. 5. Averaged single-cell Raman spectra of *R. eutropha*-GR-Mtr and *R. eutropha*-GR cultured with 30% D₂O and CO₂ under light and dark conditions.

4. Furthermore, the chain of electron/hydrogen carriers that has been created implies the involvement of non-physiologically large redox gaps, so that long-term functioning would be seriously endangered by damage due to ROS formation.

Response: Reactive oxygen species (ROS) pose a potential threat in photo-electrosynthesis as they can be toxic to cells ^[1, 2]. In this study, one possible source of ROS could be the reverse electron transfer of NADH dehydrogenase, which reduces NAD⁺ to NADH in the presence of oxygen ^[3]. To avoid the risk of ROS generation, we used a dual-chamber electrochemical reactor separated by a Nafion proton exchange membrane (PEM) to reduce the effect of the anodic oxygen generation on microbial carbon fixation in the cathodic chamber. This design ensures that oxygen produced from water splitting in the anodic chamber cannot permeate the PEM, thus preventing its entry into the cathodic chamber and the generation of ROS. In addition, the syngas we pumped into the cathode chamber consists of N₂/CO₂ without oxygen, maintaining an anaerobic condition. Although the anaerobic cathode chamber can effectively avoid ROS generation, it could sacrifice the cellular energy generated via the aerobic respiration pathway.

The goal of this study is to establish a novel hybrid system, serving as a model platform for artificial photosynthesis. Our future work will focus on achieving a balance between reducing ROS effects and improving energy generation. In fact, there are some strategies to reduce ROS when bacteria are exposed to the risk of ROS. For example, engineering bacteria to express lycopene has been shown to improve microbial resistance toward ROS ^[4]. Furthermore, the addition of antioxidant, such as glutathione, has been shown effectively to reduce ROS in the system ^[5]. We have added some discussion of ROS in the revised manuscript, as follows:

“Reactive oxygen species (ROS) is a potential concern in photo-electrosynthesis because of their toxicity to cells. In this study, a high proton motive force and reverse electron transfer could be possible factors contributing to ROS generation. To avoid the risk of ROS generation, we employed a dual-chamber electrochemical reactor with a Nafion proton exchange membrane (PEM) to separate anodic oxygen generation from microbial carbon fixation in the cathodic chamber. This design ensures that oxygen produced from water splitting in the anodic chamber cannot permeate the PEM, thus preventing its entry into the cathode chamber to generate ROS. In addition, the N₂/CO₂ syngas was continuously pumped into the cathode chamber to maintain an anaerobic condition. Although the anaerobic cathode chamber can effectively avoid ROS generation, it could sacrifice the cellular energy generated via the aerobic respiration pathway. Therefore, a strategic approach might involve controlling oxygen level while simultaneously enhancing microbial resistance to ROS. Achieving a balance between reducing ROS effects and improving energy generation would be important for the futural scaling-up and long-term operation of the system. There are some strategies to reduce ROS when bacteria are exposed to the risk of ROS. For example, engineering bacteria to express lycopene has been shown to improve microbial resistance toward ROS ^[4]. The addition of antioxidant, such as glutathione, has been shown effectively to reduce ROS in the system ^[5].”

[1] Li, Han, et al. "Integrated electromicrobial conversion of CO₂ to higher alcohols." *Science* 335.6076 (2012): 1596-1596.

[2] Torella, Joseph P., et al. "Efficient solar-to-fuels production from a hybrid microbial–water-splitting catalyst system." *Proceedings of the National Academy of Sciences* 112.8 (2015): 2337-2342.

[3] Scialò, Filippo, Daniel J. Fernández-Ayala, and Alberto Sanz. "Role of

mitochondrial reverse electron transport in ROS signaling: potential roles in health and disease." *Frontiers in physiology* 8 (2017): 428.

[4] Wu, Haoliang, et al. "Efficient production of lycopene from CO₂ via microbial electrosynthesis." *Chemical Engineering Journal* 430 (2022): 132943.

[5] Na, Yoon-Ah, et al. "Growth retardation of *Escherichia coli* by artificial increase of intracellular ATP." *Journal of Industrial Microbiology and Biotechnology* 42.6 (2015): 915-924.

A few additional minor points of criticism:

5. Line ... : we also have the bilins and the carotenoids for this

Response: We assume that reviewer is referring to Line 228, where we stated "... typical bands of canthaxanthin ..." whereas the Raman bands might also indicate bilins and carotenoids. We have revised the sentence to make it clearer as follows: "Single-cell Raman analysis showed bands at 1005, 1155 and 1517 cm⁻¹ in the engineered cells when canthaxanthin was introduced, in contrast to the cells without canthaxanthin, and these bands are consistent with the Raman spectra of pure canthaxanthin chemical (Fig. 4c and Supplementary fig. 7)."

6. Line 239: The *can* enzyme is not driving the direction of the reaction; it is only catalyzing it

Response: We have revised the sentence to make it more accurate. We have replaced "abundant *can* is expected to drive HCO₃/CO₂ balance towards CO₂" with "the overexpression of *can* is expected to accelerate the interconversion HCO₃ between CO₂".

7. Line 248: Many phototrophic bacteria have doubling times below 10 or even 5 hrs; the example cited is far from typical.

Response: We chose to compare the engineered *Ralstonia eutropha* with *Rhodospseudomonas palustris* rather than other phototrophic bacteria due to several similarities between the two species: 1) Both of them are phototrophic; however, their photosystems are unable to solely drive CO₂ fixation due to the absence of photosystem II, which is essential for water splitting and electron generation; 2) Both are able to grow on CO₂ by harvesting light energy and using an electrode as the electron source; 3) The doubling time for comparison was conducted under the photo-electrochemical conditions in an anaerobic chamber using light and an electrode as the energy source.

The reasoning for using *R. palustris* as a comparison has been added to discussion as follows:

"The photo-electrotrophic *R. eutropha* engineered in this study is similar to the anoxygenic phototrophs *Rhodospseudomonas palustris* which is a model bacterium for studying phototrophic extracellular electron uptake. Both the genetically engineered *R. eutropha* and the native *R. palustris* are phototrophic; however, their photosystems are unable to solely drive CO₂ fixation due to the absence of photosystem II which is essential for water splitting and electron generation. In the presence of an electrode, both microorganisms are able to harvest electrode-supplied electrons to drive CO₂ fixation pathway in the light. However, the underlying mechanisms of these microbial systems are different. The engineered *R. eutropha* employs rhodopsin to generate proton motive force without involving electron transfer. We used the synthetic design to couple proton movement with electron transfer to generate the energy required for CO₂ fixation. In contrast, *R. palustris*'s photosystem P₈₇₀ naturally involves electron transfer and can excite electrons in the light. Although

the synthetic design showed comparable performance in supporting photo-electroautotrophic growth, further investigations are needed to deepen our understanding of the photosynthetic electron transport chain.”

8. Line 258: “therefore contributing to sustainable biomanufacturing”: overly optimistic as long as an MCP of the product is not provided.

Response: Based on the findings of this study, it suggests that this system has the potential to be extended for the transformation of other non-photosynthetic bacteria into photosynthetic ones. Given this, we have removed the phrase “contributing to sustainable biomanufacturing” to ensure our claims more accurately with the available information.

9. Line 282: “in contrast to the efficiency of natural photosynthesis (~1%)” – why are the authors so pessimistic about natural photosynthesis?

Response: We have replaced it with “This efficiency surpasses the photosynthesis efficiency for most of natural photosynthesis plant systems (~1%) and comparable to solar-to-biomass yields observed over a growing season for domestic C3 and C4 crops (2.9–4.3%)^[1]”

[1] Blankenship, Robert E., et al. "Comparing photosynthetic and photovoltaic efficiencies and recognizing the potential for improvement." *science* 332.6031 (2011): 805-809.

10. Line 288: “consortia for maximizing the light-harvesting spectrum, increasing the overall light energy transfer efficiency” – it is important to realize that although chlorophyll-based photosynthesis shows increased absorbance in certain wavelength regions, there is no part of the PAR region where there is no absorption. Hence it is straightforward to grow these organisms so that they absorb all available PAR light.

Response: Thank you for the comment. Chlorophyll indeed exhibits varying degrees of absorbance across the PAR spectrum. We have revised the sentence “Chlorophyll does not absorb photons in the photosynthetically available radiant (PAR) waveband evenly, with only a low degree of absorbance occurring in the green region^[1]. In contrast, rhodopsin strongly absorbs green-blue light (~500 nm). The complementary light absorption offers the possibility of integrating chlorophylls and rhodopsin photosystems in either a pure culture^[2] or a mixed community^[3] to maximise the utilisation of the solar energy.”

[1] Kume, Atsushi. "Importance of the green color, absorption gradient, and spectral absorption of chloroplasts for the radiative energy balance of leaves." *Journal of plant research* 130 (2017): 501-514.

[2] Chen, Que, et al. "Combining retinal-based and chlorophyll-based (oxygenic) photosynthesis: Proteorhodopsin expression increases growth rate and fitness of a Δ PSI strain of *Synechocystis* sp. PCC6803." *Metabolic engineering* 52 (2019): 68-76.

[3] Gómez-Consarnau, Laura, et al. "Microbial rhodopsins are major contributors to the solar energy captured in the sea." *Science advances* 5.8 (2019): eaaw8855.

11. Line 307: “Therefore, our demonstration of artificial photosynthesis in *R. eutropha* could unveil the presence of extensive yet undocumented rhodopsin phototrophs in oceanic environments” – for me this is too much speculation; finding such organisms via gene probes to me seems like a much more rational approach.

Response: Thank you for the suggestion. We have revised the sentence to “Therefore, our demonstration of artificial photosynthesis in *R. eutropha* could guide future research into connection between rhodopsin photosystems and extracellular

electron transfer systems at the gene level in oceanic environments.”

12. Line 353: “Flavin mononucleotide (FMN) was also added to be anchored on the Mtr with a concentration of 10 $\mu\text{mol/L}$ ” – please explain what ‘anchored’ means here.

Response: “anchored” means that FMN as a cofactor can bind strongly to the outer membrane cytochrome Mtr. To make it clearer, we have revised the sentence to “Flavin mononucleotide (FMN) was added at a final concentration of 10 $\mu\text{mol/L}$ to investigate the interaction between outer-membrane cytochrome Mtr and FMN.

13. Line 437: Please explain: “5 m of white LED light strips were set adjacent to the working chamber”.

Response: To provide even illumination across the reactor during the experiment, we positioned white LED light strips directly around the exterior of the working chamber. We have revised the sentence to “5 meters of white LED light strips encircled the exterior of the working chamber to ensure even light distribution during the experiment.”

14. Line 453: “to exhaust any potential internal electron storage, such as PHB” – it is not sufficiently clear where the mutant strain was used that could not synthesize PHB.

Response: We used regular *R. eutropha* strains without deletion of PHB synthetic pathway in the characterisation experiments such as nitrate reduction and incubation with heavy water D_2O . For biomass growth experiments, we used *R. eutropha* Δpha (RHM5) mutant. We have highlighted the use of *R. eutropha* Δpha (RHM5) strain in the “A hybrid system for light-driven CO_2 fixation into biomass” section of Materials and Methods, mentioning that “..., the RHM5 strain, an *R. eutropha* mutant without PHB biosynthetic pathways was used for biomass growth”.

15. Line 491: “10 $\mu\text{mol/L}$ of flavin was added at an interval of 24 h to reduce their photochemistry”. Please explain what “their” refers to in this sentence.

Response: ‘their’ refers to the flavin molecules themselves. We have replaced “their” with “flavin’s”.

Reviewer #2 (Remarks to the Author):

Summary:

In this manuscript, Tu et al. describe their engineering of *Ralstonia eutropha* for growth on CO₂ and a cathode, using light as a supplemental energy source. *R. eutropha* is capable of carbon fixation but cannot naturally interact with electrodes or use light energy, so a transmembrane electron conduit (Mtr) from *Shewanella oneidensis* and a rhodopsin from *Gloeobacter* (GR) were heterologously expressed. The manuscript presents evidence for functional expression of both Mtr and GR. The engineered strain was capable of growth using only light, the cathode, and CO₂, demonstrating a significant advance for microbial electrosynthesis. Further strain improvements were also made, including adding a carbonic anhydrase and removing the pathway for PHB production.

Response: Thank you for your suggestions on this manuscript. A point-by-point response on all issues raised is provided below.

Major comments:

1. Can the authors speculate on the identity of the inner membrane proteins that transfer electrons from MtrA to the quinone pool in *R. eutropha*? It would also be useful to note the types of NADH dehydrogenases present in *R. eutropha* and speculate on which is/are involved in inward electron transfer.

Response: A series of inner membrane enzymes in electron transport chain could be associated with electron transfer. We speculate that nitrate reductase and hydrogenase in *R. eutropha* could play an important role in the inward electron transfer.

a) The nitrate reductase NapC was reported to functionally replace CymA which is able to obtain electrons from Mtr pathway^[1, 2]. There is also evidence indicating that nitrate reductase was also upregulated during the processes of extracellular electron transfer in *R. eutropha*^[3].

b) Another potential candidate is the membrane-bound hydrogenase of *R. eutropha*, which can transfer electrons to the quinone pool. This mechanism is important in chemoautotrophic metabolism. In *Shewanella* system, Mtr was reported to pass electrons onto hydrogenase for redox reactions^[4]. Therefore, it is also reasonable to speculate that hydrogenase in *R. eutropha* could also contribute to inward electron transfer from MtrA to quinone pool.

We added the discussion of the possible enzymes for electron transfer in Discussion as follows.

“In the model electroactive bacterium *S. oneidensis* MR-1, the outer-membrane MtrCAB pathway is usually coupled with an inner-membrane protein CymA to involve electron transfer. However, a recent study on the engineered *E. coli* with a Mtr pathway showed that CymA might not be necessary for inward electron transport. Several other inner-membrane enzymes such as NapC can function as CymA. In addition, a recent study on *S. oneidensis* MR-1 reported hydrogenase can obtain electrons from Mtr pathway. Inspired by the finding in *S. oneidensis* MR-1, we propose that, in the native system of *R. eutropha* the periplasm-facing membrane-bound hydrogenase could also play an important role in Mtr-mediated inward electron transfer.”

There are two types of NADH dehydrogenases present in *R. eutropha*, which are *nuo* and *ndh*. According to a previous study on *S. oneidensis*, *nuo* is critical to inward electron transfer from the electrode for NADH generation [5]. Therefore, we propose the *nuo* in *R. eutropha* could also play an important role in inward electron transfer.

We also added the following to the Discussion Section as follows:

“In *R. eutropha*, there are two types of NADH dehydrogenase, namely *nuo* and *ndh*. *nuo* is a proton-translocating protein while *ndh* is a non-proton pumping membrane-bound protein. According to a previous study on *S. oneidensis*, *nuo* is critical to inward electron transfer from the electrode for NADH generation. In this system, the NADH generation was impeded after the addition of cyanide m-chlorophenyl hydrazone (CCCP), a protonophore (Supplementary fig. 11). Therefore, it is reasonable to infer the significance of the proton-translocating *nuo* in *R. eutropha* in the process of inward electron transfer for NADH generation.”

[1] Feng, Jiao, et al. "Direct electron uptake from a cathode using the inward Mtr pathway in Escherichia coli." *Bioelectrochemistry* 134 (2020): 107498.

[2] Baruch, Moshe, et al. "Electronic control of redox reactions inside Escherichia coli using a genetic module." *PloS one* 16.11 (2021): e0258380.

[3] Nishio, Koichi, et al. "Extracellular electron transfer enhances polyhydroxybutyrate productivity in *Ralstonia eutropha*." *Environmental Science & Technology Letters* 1.1 (2014): 40-43.

[4] Han, He-Xing, et al. "Reversing electron transfer chain for light-driven hydrogen production in biotic–abiotic hybrid systems." *Journal of the American Chemical Society* 144.14 (2022): 6434-6441.

[5] Tefft, Nicholas M., Kathryn Ford, and Michaela A. TerAvest. "NADH dehydrogenases drive inward electron transfer in *Shewanella oneidensis* MR-1." *Microbial Biotechnology* 16.3 (2023): 560-568.

2. The engineered metabolism appears pretty similar to what has been reported for *Rhodopseudomonas palustris*. While one study on this organism was cited in the discussion, I think it would be helpful to also mention *R. palustris* in the introduction and describe the similarities and differences between it and the engineered system.

Response: Thank you for the suggestions. We have added description about the similarities and differences between our system and *Rhodopseudomonas palustris* in Introduction and Discussion.

Introduction: “..... In the model photosynthetic bacterium *Rhodopseudomonas palustris*, there are cytochromes presented on the outer membrane to enable the electron transfer between intracellular and extracellular electron acceptors and electrons. As with *R. palustris*, the proton gradient not only drives ATP production via ATP synthase but also plays a key role in reversing the function of proton-translocating NADH dehydrogenase for regeneration of reducing equivalents.”

Discussion: “The photo-electrotrophic *R. eutropha* engineered in this study is similar to the anoxygenic phototroph *Rhodopseudomonas palustris* which is a model microbial system for studying phototrophic extracellular electron uptake. Both the genetically engineered *R. eutropha* and the native *R. palustris* are phototrophic; however, their photosystems are unable to solely drive CO₂ fixation due to the absence of photosystem II which is essential for water splitting and electron generation. In the presence of an electrode, both microorganisms are able to harvest electrode-supplied electrons to drive CO₂ fixation pathway in the light. However, the underlying mechanisms of these microbial systems are different. The engineered *R.*

eutropha employs rhodopsin to generate proton motive force without involving electron transfer. We used the synthetic design to couple proton movement with electron transfer to generate the energy required for CO₂ fixation. In contrast, *R. palustris*'s photosystem P₈₇₀ naturally involves electron transfer and can excite electrons in the light. Although the synthetic design showed comparable performance in supporting photo-electroautotrophic growth, further investigations are needed to deepen our understanding of the photosynthetic electron transport chain.”

3. The representations of the transhydrogenase and the NADH dehydrogenase in figures 3 and 4 seem somewhat misleading. Since both of these couple electron transfer and proton transfer, I think they should be represented as spanning the entire membrane.

Response: Thank you for the suggestions. We have now revised the representation of transhydrogenase and the NADH dehydrogenase in figure 2 and 4, to reflect their spanning the entire membrane.

Minor comments:

4. The references should be checked for correct italicization and capitalization.

Response: Done. All bacterial species have been italicized, and the first letters of the titles have been capitalized.

5. The 'c' in cytochrome c should be italicized throughout.

Response: Done.

6. Line 19: change 'out' to 'outer'

Response: Done.

7. Line 72: change 'heterogeneously' to 'heterologously'

Response: Done.

8. Line 103: I think this should be 'inset' instead of 'inlet'. There are also other instances

Response: Thanks. We have corrected them.

9. Line 323: remove 'as'

Response: Done.

10. Line 432: indicate supplier and catalog number for the carbon cloth

Response: Thank you for your suggestion. We have added more information for carbon cloth.

11. Line 434/435: indicate initial pH

Response: Done.

12. Supplementary line 39: change 'streains' to 'strains'

Response: Done.

Reviewer #3 (Remarks to the Author):

The authors engineer an artificial pathway for light-driven CO₂ fixation in *R. eutropha*. Heterologously expressed rhodopsin is used to create a proton gradient that, when combined with electron flow from heterologously expressed MtrABC, reverses the NADH dehydrogenase activity. The generated NADH is converted to NADPH to promote CO₂ fixation via the Calvin cycle. The authors further enhance light-harvesting and electron transfer through canthaxanthin and flavin addition, respectively. Enhanced carbonate to CO₂ conversion was also achieved through carbonic anhydrase expression. The authors rely on Raman measurements to confirm protein expression and enhanced metabolic activity through C13 and deuterium-labeling.

Response: Thanks for the reviewer's thorough summary.

Previous studies have enhanced NADH production through rhodopsin expression in *R. eutropha* (using a mediator for electron transfer for carbon fixation applications, Huang et al.) and in *S. oneidensis* (using its native MtrCAB pathway for electron transfer, TerAvest et al.). The novelty in this work is the co-expression of the MtrCAB pathway in the rhodopsin-engineered *R. eutropha*. To my knowledge, the MtrCAB pathway has not been previously expressed in *R. eutropha*, and the application of the MtrCAB pathway for CO₂ fixation in any host has yet to be demonstrated. This study thus opens the doors to an impactful and previously unexplored application of this pathway which I believe aligns well with the high standards of this journal. However, the Raman analysis does not provide a convincing basis for supporting the conclusions of this study, and the authors are missing comparisons between their new strain and the previous strain without the MtrCAB pathway. I therefore recommend publishing this manuscript with major revisions.

Response: We are grateful to the Reviewer for taking the time to improve our manuscript. A point-by-point response on all issues raised is provided below.

Major:

1. SDS-PAGE is needed to confirm correct protein size and localization of heterologous proteins, as well as heme insertion for the MtrCAB complex.

Response: We thank the reviewer's suggestions. In response, we have carried out an additional experiment and added gel electrophoresis data of proteins to validate the expression of Mtr proteins. Whole-cell proteins were extracted and subsequently separated on a 10% Bis-Tris Gel. The 3,3',5,5'-tetramethylbenzidine (TMBZ) peroxidase stain method was employed to identify cytochromes *c*. Notably, the protein sizes observed in the gel align with previous studies^[1, 2] (MtrA: ~35kDa, MtrC: ~70kDa), and the picture for the protein gel has been included in the supplementary materials (Supplementary fig. 3). We added the result to the revised manuscript: "Proteins from whole cell extracts were separated by SDS-PAGE, and heme *c* containing proteins MtrC and MtrA were identified by heme staining (Supplementary fig. 3)." The confirmation of Mtr pathway now includes visual assessment of Mtr-complex expression through changes in cell pellet colour, the presence of heme Raman bands associated with cytochromes, SDS-PAGE gel analysis, correctly sized protein bands, functional validation through nitrate reduction experiments, D₂O experiments, and growth tests. The materials and methods have been added to the supplementary information.

Supplementary fig. 3. Heme staining of whole cell lysates of the *R. eutropha*-Mtr at arabinose inducer concentrations of 0%, 0.02%, and 0.1%.

[1] Jensen, Heather M., et al. "CymA and exogenous flavins improve extracellular electron transfer and couple it to cell growth in Mtr-expressing *Escherichia coli*." *ACS synthetic biology* 5.7 (2016): 679-688. (Figure 1b)

[2] Feng, Jiao, et al. "Direct electron uptake from a cathode using the inward Mtr pathway in *Escherichia coli*." *Bioelectrochemistry* 134 (2020): 107498. (Figure S2)

2. Raman analysis has a high signal to noise ratio. The authors should corroborate their measurements (including protein expression and isotope labeling) with more sensitive techniques.

Response: The reviewer is correct that Raman scattering signal is intrinsically weak. Single-cell Raman spectroscopy in this study has been used to characterise biomolecules of cytochromes, rhodopsin, and canthaxanthin-GR as well as C–D band and phenylalanine band for stable isotope probing. This study employed a green incident laser light with a wavelength of 532 nm for our Raman measurements. Importantly, all the biomolecules we characterized with Raman absorb visible light at wavelengths very close to the incident light. For example, cytochromes absorb light at 530 nm ^[1], rhodopsin at 520 nm ^[2], and canthaxanthin-GR at 485 nm ^[3]. This proximity in wavelengths results in a resonance effect, which significantly enhances the Raman scattering cross-section and consequently improves the signal-to-noise ratios by orders of magnitude. Furthermore, the phenylalanine band we selected for ¹³C-labelling experiments exhibits an aromatic ring that also contributes to this resonance effect. Regarding our use of Raman-deuterated water, we would like to emphasize that this technique has been widely applied in probing microbial activities ^[4] and offers a unique, universally applicable approach for non-destructively

measuring single-cell activities^[5].

We believe that our choices for Raman characterization in this study are justified due to either the resonance effect significantly enhancing the signal or the unique advantages of the technique, particularly at the single-cell level. To further clarify our rationale and methodology, we have incorporated additional discussion in the methodology section dedicated to Raman techniques.

Lastly, in response to the reviewer's point about the expression specificity of the MtrCAB heme complex, we appreciate your suggestion. As mentioned in the last response, we have conducted SDS-PAGE experiments to characterize the expression of this protein. This information has been included in the manuscript to provide a more comprehensive overview of our experimental approach.

[1] Okada, M. et al. Label-free Raman observation of cytochrome c dynamics during apoptosis. *Proc Natl Acad Sci U S A* 109, 28–32 (2012).

[2] Song, Y. et al. Proteorhodopsin Overproduction Enhances the Long-Term Viability of *Escherichia coli*. *Appl Environ Microbiol* 86, (2019).

[3] Chuon, K. et al. The role of carotenoids in proton-pumping rhodopsin as a primitive solar energy conversion system. *Journal of Photochemistry and Photobiology B: Biology* 221, 112241 (2021).

[4] Wang, Y., Huang, W. E., Cui, L. & Wagner, M. Single cell stable isotope probing in microbiology using Raman microspectroscopy. *Curr. Opin. Biotechnol.* 41, 34–42 (2016).

[5] Berry, D. et al. Tracking heavy water (D₂O) incorporation for identifying and sorting active microbial cells. *Proc. Natl. Acad. Sci. U.S.A.* 112, E194-203 (2015).

3. The MtrCAB complex is not significantly effective in promoting charge transfer in the absence of inner membrane (CymA) or periplasmic (STC) expression (see Jensen et al. DOI: 10.1021/acssynbio.5b00279; Mouhib et al. DOI: 10.1101/2021.08.28.458029). It is unclear if this strain can outperform the previous rhodopsin strain without the MtrCAB expression, especially since this strain does not have a pathway for transfer from the outer membrane. The performance of this strain needs to be compared to the previous rhodopsin strain without MtrCAB (with and without a mediator).

Response: According to previous studies that MtrCAB-mediated inward electron transfer was not significantly influenced by inner membrane (CymA) or periplasmic (STC) expression, unlike outward electron transfer^[1, 2]. In 2010, Jensen et al.^[3] reported NapC in *E. coli* could function as CymA in *S. oneidensis* to help outward electron transfer, although the electron transfer rate was limited. In 2016, Jensen et al.^[4] optimised the electron transfer pathway by adding CymA with MtrCAB to enhance outward electron transfer in *E. coli*. In 2021, Baruch et al.^[1] from the same group reported that CymA might not be the limiting factor for inward electron transfer because they found *E. coli*-Mtr and the *E. coli*-CymAMtr strains have similar current consumption. According to the previous study and considering the rich redox membrane-bound proteins in *R. eutropha*, we only introduced MtrCAB to the system, and our electrochemical nitrate reduction experiments showed that combination of Mtr pathway and native electron transport chain can effectively transport electron onto electron transport chain; therefore, we used *R. eutropha*-Mtr without additional

expression of CymA in the rest of our experiments. We added more discussion to the revised manuscript as follows:

“In the process of electron transfer in the engineered *R. eutropha*, the native inside biomolecule plays an important role in obtaining electrons from Mtr pathway. In the model electroactive bacterium *S. oneidensis* MR-1, the outer-membrane MtrCAB pathway is usually coupled with an inner-membrane protein CymA to involve electron transfer. However, a recent study on the engineered *E. coli* with a Mtr pathway showed that CymA might not be necessary for inward electron transport. Several other inner-membrane enzymes such as NapC can function as CymA. In addition, a recent study on *S. oneidensis* MR-1 reported hydrogenase can obtain electrons from Mtr pathway. Inspired by the finding in *S. oneidensis* MR-1, we proposed that, in the native system of *R. eutropha* the periplasm-facing membrane-bound hydrogenase could also play an important role in Mtr-mediated inward electron transfer.”

We have also added the additional experimental result as supplementary fig. 8 to compare the initial and final optical density (OD₆₀₀) in 5 days of incubation under the photoelectrochemical system for various strains, including those with and without Mtr pathway, as well as those with and without an electron mediator.

Supplementary fig. 8. Initial and final optical density (OD₆₀₀) for the *R. eutropha* RHM5-GR-Mtr strains with and without induction Mtr pathway after 5 days incubations with and without 50 μ M FMN as the mediator (**: $p < 0.01$, ***: $p < 0.005$, n.s.: not significant, data are means \pm SD, $n = 3$).

[1] Baruch, Moshe, et al. "Electronic control of redox reactions inside Escherichia coli using a genetic module." PloS one 16.11 (2021): e0258380.

[2] Feng, Jiao, et al. "Direct electron uptake from a cathode using the inward Mtr pathway in Escherichia coli." Bioelectrochemistry 134 (2020): 107498.

[3] Jensen, Heather M., et al. "Engineering of a synthetic electron conduit in living cells." *Proceedings of the National Academy of Sciences* 107.45 (2010): 19213-19218.

[4] Jensen, Heather M., et al. "CymA and exogenous flavins improve extracellular electron transfer and couple it to cell growth in Mtr-expressing *Escherichia coli*." *ACS synthetic biology* 5.7 (2016): 679-688.

Minor:

4. The text should be reviewed to make sentences more concise and easier to read. Some examples:

- "Microbial rhodopsins as the most widespread phototrophic system throughout the microbial world are hypothesised to have a significant role in microbial carbon fixation which is beyond our current knowledge" to "Microbial rhodopsins, the most widespread phototrophic proteins in the microbial world, are hypothesised to have a significant role in microbial carbon fixation."

Response: Thanks for your suggestion. We have revised it.

- "...rhodopsin phototrophy was proven to generate..." – replace "proven" with "shown"

Response: Done.

- "...rhodopsin cannot solely drive the CO₂ fixation pathway by its proton transport." – replace "by" with "with"

Response: Done.

- "Such Yin (electrons) and Yang (protons) interactions drive the photochemical reactions in this artificial photosynthesis." – analogy to Yin and Yang interactions are not necessary

Response: Thanks. We have deleted "Yin" and "Yang" .

5. Relabel "Discussion" to "Conclusion"

Response: We have carefully considered this recommendation. According to brief guide for submission to *Nature communications*, it is not mandatory to include a "Conclusion" section. Given the substantial content and discussions in the existing 'Discussion' section, we think that keeping it as such is appropriate.

6. Correction of typos and improvement in figures. Some examples:

- Fig. 1 change "Nafin" to "Nafion", and label anode and cathode compartments in figure

Response: Done.

- Fig. 2 – abbreviations (IN, OM, IP) should be explained in caption, spacing before "(d)" needs to be added in caption, caption of (f) "percentage of cells" is unclear (is this the percentage in the number of cells in each culture that is expressing vs. those that aren't? how is this determined?)

Response: Thanks for your suggestions. (1) IM, OM, IP represented inner membrane, outer membrane and the inner-membrane protein for transferring electrons to electron transport chain. (2) We have added a space before (d). (3) "percentage of GR-expressing cells" is determined by examining single-cell Raman spectra of

individual cells. We calculated this percentage by dividing the number of cells exhibiting the 1530 cm^{-1} GR band by the total cell count. To make it clearer, we have added more details in the caption as follows: “The percentage of GR-expressing cells, i.e., with the 1530 cm^{-1} GR band, in induced (number of measured cells $n = 297$) and uninduced (number of measured cells $n = 306$) groups.”.

- Fig. 4 – GRCX needs to be specified

Response: Done. We have revised it to GR-canthaxanthin complex (GRCX).

Reviewer #4 (Remarks to the Author):

Summary & Noteworthy results. This work builds on a previously established strain of *Ralstonia eutropha* (*Cupriavidus necator*) H16 engineered to express rhodopsin from *Gloeobacter* (reference #10, Davison PA, et al). The primary innovation was the introduction of an extracellular electron transfer conduit known as the multi-heme transmembrane reductase (MtrCAB), which enabled faster extracellular electron transfer into *R. eutropha*'s native internal electron transfer chain (ETC) via a redox mediator (riboflavin) and external electrode. The authors claim the flow of low potential electrons into the quinone pool via the MtrCAB, coupled with the inverted proton motive force driven by the light-activated rhodopsin, drives the native ETC in reverse. The authors suggest this ultimately drives more NADH and NADPH synthesis, in a physiologically reversed manner, which is then utilized in the CBB cycle to boost CO₂ capture and biomass production. The work uses rigorous electrochemical and Raman, UV-VIS spectroscopic methods to monitor Mtr functional expression, metabolic activity, and cell growth. The method and the results are well-presented and interesting. However, I find the interpretation of the results far-fetched, and unsatisfactorily unsupported by the literature.

Response: Thanks for the comprehensive summary. We have revised the manuscript according to the reviewers' comments.

Significance to the field, comparison to established literature & Originality. The host system and its characterization with an electrode is partially original, as it builds upon a previously existing system that uses a transmembrane-mediated electron transfer that proceeds via an unclear mechanism (ref 10). Secondly, while the incorporation of extracellular ET pathways for electrode-driven biosynthesis has also been shown (ref 9: Tefft NM, TerAvest MA.), this article presents the first expression of both proteins in this particular host organism, *Ralstonia eutropha*. The functional incorporation of MtrCAB into this host is indeed an innovative step forward and will be significant for future systems incorporating such an extracellular ET pathway.

Response: Thanks for the reviewer's positive feedback and recognition of the significance of this study.

Does the work support the conclusions and claims, or is additional evidence needed? Are there any flaws in the data analysis, interpretation and conclusions? The methodology is rigorous, the presentation of the data is high quality and meets the standards of the journal, and there is enough detail provided in the methods for the work to be reproduced. Overall, the authors apply a rigorous scientific method and present their results with great clarity. However, the interpretation of the results, particularly related to the mechanistic hypotheses, are seemingly not feasible and far-fetched. The mechanism of 'artificial photosynthesis' via a pmf-driven voltage-up conversion in NAD(P)H dehydrogenase is, according to my knowledge, unprecedented. While it surely is an interesting hypothesis, I cannot rationalize a physiological basis for this seemingly complex and inverted reaction to exist, nor can I find any supporting literature outside of the authors' own previous claims. This issue prohibits publication and requires revision. I will elaborate on this issue in the text that follows.

Response: We understand the reviewer's concerns regarding the mechanism of NADH dehydrogenase and the potential association between NADH regeneration and proton motive force (PMF). In many instances, the proton-translocating NADH dehydrogenase oxidises NADH to generate a PMF for ATP production. However, NADH dehydrogenase can also function in reverse to catalyze uphill electron

transport from the quinone pool to reduce NAD⁺ to NADH in the anoxygenic phototrophs *Rhodopseudomonas palustris* ^[1], *Rhodobacter capsulatus* ^[2], and *Rhodobacter sphaeroides* ^[3]. The central concern here is whether the reversing function of NADH dehydrogenase consumes PMF, or if a higher PMF level is more favourable for generating NADH.

To address these concerns, we have conducted additional experiments and consulted relevant literature. We have also revised our manuscript to provide a clearer explanation for our claims. A point-by-point response on all issues raised is provided below.

[1] Guzman, Michael S., et al. "Phototrophic extracellular electron uptake is linked to carbon dioxide fixation in the bacterium *Rhodopseudomonas palustris*." *Nature communications* 10.1 (2019): 1355.

[2] Herter, Stefan Michael, Christiane Maria Kortlüke, and Gerhart Drews. "Complex I of *Rhodobacter capsulatus* and its role in reverted electron transport." *Archives of microbiology* 169 (1998): 98-105.

[3] Spero, Melanie A., et al. "Different functions of phylogenetically distinct bacterial complex I isozymes." *Journal of Bacteriology* 198.8 (2016): 1268-1280.

Major issue: The interpretation of increased biomass production via voltage upconversion via a pmf.

1. How does the quinone inject an electron into the NAD(P)H dehydrogenase? It is a ~250 mV thermodynamically unfavorable jump, according to Figure 5. The authors have shown the midpoint potential of the Quinone (-0.08V). However, I would assume the effective Quinone/Quinol potential to be much more negative, as the quinone pool is largely reduced by the MtrCAB supplied low potential electron flux. Perhaps that could explain why ET is injected into NAD(P)H dehydrogenase.

Response: NADH dehydrogenase reduced NAD⁺ to NADH with electrons received from quinol pool via reverse electron transfer. There are two classes of NADH dehydrogenase in bacteria: the proton- or sodium-pumping NADH-1 enzyme complex (e.g., *nuo*) and NADH-2 which is a nonproton-translocating enzyme complex, such as *ndh*. The proton-translocating NADH dehydrogenase is are critical to inward electron transfer from the electrode for NADH generation ^[1]. The reversing function of NADH dehydrogenase to regenerate NADH requires 1) a highly reduced Q-pool ([QH₂] > [Q]) to provide the electrons and 2) a high proton gradient to drive protons back through the complex in the opposite direction to forward catalysis ^[2]. In *Shewanella oneidensis* system, Rowe et al. ^[3] reported that PMF might aid in reverse electron flow to regenerate NADH via NADH dehydrogenase *nuo*. While Rowe et al.'s study did not introduce rhodopsin to power PMF generation like ours, they used O₂ as the electron acceptor which can generate PMF via *S. oneidensis*'s respiration chain. In addition, Tefft et al. ^[1] reported that inhibition of PMF by CCCP (carbonyl cyanide m-chlorophenyl hydrazine, a proton motive force inhibitor) stopped the reduction reaction in *S. oneidensis*, which could also indicate the potential connection between PMF and reversible NADH synthesis by NADH dehydrogenase.

In addition, we cited some papers to address the point. The reduction potential of the quinone ranges from -0.1V ^[4] to -0.267V ^[5] vs NHE, which could still be difficult to reduce NAD⁺ (-0.32V) without the support of PMF. While the midpoint potential of the quinone (c.a. -0.08V) may not represent the effective potential, we maintain consistency with the labelling of enzymes and mediators in chlorophyll-based photosystems by using midpoint potential. We have added text in the Discussion section to emphasize the more negative potential of the quinone for NAD⁺ reduction, as follows:

"The reversing function of NADH dehydrogenase is a key to regenerating NADH. The reverse electron transfer from the quinol pool (with a midpoint potential, E_m , approximately -0.08 V) to NAD^+ ($E_m = -0.32$ V) is thermodynamically unfavourable (Fig. 5). The reversed process requires a highly reduced quinol pool to supply electrons and a high proton motive force to drive protons back through the NADH dehydrogenase in the reverse direction to forward catalysis. In this study, electrons flow inwardly from the external electrode to Mtr pathway then leading to the reduction of quinol pool. The effective potential of the highly reduced quinol pool should be more negative than its electrochemical midpoint potential and approached NADH potential. In the meantime, the proton motive force generated by rhodopsin facilitated the NADH dehydrogenase-mediated reverse electron flow for NADH regeneration."

[1] Tefft, Nicholas M., Kathryn Ford, and Michaela A. TerAvest. "NADH dehydrogenases drive inward electron transfer in *Shewanella oneidensis* MR-1." *Microbial Biotechnology* 16.3 (2023): 560-568.

[2] Wright, John J., et al. "Reverse electron transfer by respiratory complex I catalysed in a modular proteoliposome system." *Journal of the American Chemical Society* 144.15 (2022): 6791-6801.

[3] Rowe, Annette R., et al. "Tracking electron uptake from a cathode into *Shewanella* cells: implications for energy acquisition from solid-substrate electron donors." *MBio* 9.1 (2018): 10-1128.

[4] Dharmaraj, Karuppasamy, et al. "The acid-base and redox properties of menaquinone MK-4, MK-7, and MK-9 (vitamin K2) in DMPC monolayers on mercury." *European Biophysics Journal* 49.3-4 (2020): 279-288.

[5] Komlódi, Tímea, et al. "Coupling and pathway control of coenzyme Q redox state and respiration in isolated mitochondria." *Bioenergetics Communications* 2021 (2021): 3-3.

2. How does a proton gradient drive NADH reduction? I cannot rationalize how the light-activated GR transfer of protons into the lumen will drive NAD^+ to NADH reduction (which is physiologically reversed). In fact, removing the protons from that side of the membrane should make $\text{NAD}^+ + \text{H}^+ + \text{e}^- \rightarrow \text{NADH}$ even more difficult, as protons are being depleted from this reaction. The authors must rationalize this mechanism or provide an alternative hypothesis before I can accept this manuscript. For instance, can the authors exclude the hypothesis that the light-driven GR-proton pump is simply driving a pmf that drives more ATP synthesis, and the additional electrons from the MtrCAB are helping drive electrons into NAD(P)H dehydrogenases?

Response: Proton-translocating NADH dehydrogenase is a crucial NADH-quinone oxidoreductase that spans the entire cellular inner membrane. Its role in both NADH oxidation and reduction is important for biological metabolism. When catalysing the oxidation of NADH, NADH dehydrogenase pumps protons outward, generating proton gradient to power ATP synthesis. In its reversing function, NADH dehydrogenase catalyses the reduction of NAD^+ to NADH in which a high proton motive force is favourable for enhancing this process^[1].

To illustrate this mechanism, consider the model photosynthetic and electroactive bacterium *Rhodospseudomonas palustris*, in which the proton motive force is important for extracellular electron uptake and NADH generation. In the presence of light^[2], the excitation of its photosystem P_{870} leads to the transfer of electrons to quinol in a thermodynamically favourable process, accompanied by the generation of a proton gradient. The proton gradient further supports the reversing function of

NADH dehydrogenase [2]. Even in dark conditions [3], *R. palustris* can also obtain electrons from an electrode to generate NADH in the presence of an electron acceptor (e.g., nitrate). The presence of an electron acceptor triggers the generation of a proton motive force through the respiration pathway and then support NADH generation. In summary, the proton motive force is important for the regeneration of NADH via NADH dehydrogenase. In our system, we did not introduce an exogenous electron acceptor. Instead, we relied on the proton motive force generated by rhodopsin to power NADH regeneration.

In our biomass growth experiments, the *R. eutropha* RHM5-GR-Mtr did not show significant growth in the dark, indicating the importance of rhodopsin to drive CO₂ fixation. We acknowledge that this result may not directly reflect that the proton motive force can be used to drive reverse electron flow from quinol to NADH dehydrogenase to reduce NAD⁺. Therefore, to address the reviewer's concern, we conducted an additional experiment to compare the NADH/NAD⁺ levels with and without addition of CCCP, a proton motive force inhibitor. The results showed that the CCCP-treated group showed lower NADH/NAD⁺ levels compared to the group without, which could indicate the link between the proton motive force and NADH regeneration in this system. The new data has been added into supplementary as supplementary fig. 11.

Supplementary fig. 11. NADH/NAD⁺ ratios in the *R. eutropha*-GR-Mtr under photoelectrosynthetic conditions with and without carbonyl cyanide m-chlorophenyl hydrazone (CCCP) treatment. Statistics were performed with Student's t-test (*: $p < 0.05$, data are means \pm SD, $n = 3$).

We have also added some discussions to the revised manuscript as follows: "In *R. eutropha*, there are two types of NADH dehydrogenase, namely *nuo* and *ndh*. *nuo* is a proton-translocating protein while *ndh* is a non-proton pumping membrane-bound protein. According to a previous study on *S. oneidensis*, *nuo* is critical to inward electron transfer from the electrode for NADH generation. In this system, the NADH generation was impeded after the addition of cyanide m-chlorophenyl hydrazone (CCCP), a protonophore (Supplementary fig. 11). Therefore, it is reasonable to infer the significance of the proton-translocating *nuo* in *R. eutropha* in the process of inward electron transfer for NADH generation."

The methods have been added to the materials and methods section as follows: "..... To investigate the effect of the proton motive force on the reversing

function of NADH dehydrogenase, the induced *R. eutropha*-GR-Mtr strains were incubated in the photoelectrochemical system with illumination for 24 hours and then 100 μM of the proton motive force inhibitor, carbonyl cyanide *m*-chlorophenyl hydrazone (CCCP) was injected to the working chamber for additional 24-hour incubation. NADH/NAD⁺ ratios were compared between the CCCP-treated group and the control group with the blank solvent dimethyl sulfoxide."

[1] Brandt, Ulrich. "Energy converting NADH: quinone oxidoreductase (complex I)." *Annu. Rev. Biochem.* 75 (2006): 69-92.

[2] Guzman, Michael S., et al. "Phototrophic extracellular electron uptake is linked to carbon dioxide fixation in the bacterium *Rhodopseudomonas palustris*." *Nature communications* 10.1 (2019): 1355.

[3] Liu, Xing, et al. "Syntrophic interspecies electron transfer drives carbon fixation and growth by *Rhodopseudomonas palustris* under dark, anoxic conditions." *Science Advances* 7.27 (2021): eabh1852.

3. Line 136: The authors cite that previous studies that they claim 'prove' pmf driven voltage up conversion. First and foremost, in science one can never definitively 'prove' anything, you can offer the best and most well-informed explanation. Secondly, are there indications from other literature sources that support this pmf-upconversion hypothesis? Specifically, is there any evidence to suggest that NAD(P)H dehydrogenase activity is dependent upon a pmf? E.g. Are the midpoint potentials of the cofactor in NADP dehydrogenase pH dependent? For ATP synthase this is clear, the pmf and flux of protons results in conformation changes of ATP synthase that drive ATP synthesis. However, I can find no such conformational change in NAD(P)H dehydrogenase, and no support from the literature that NADH activity is pmf dependent. Thirdly, the fact that your previous mechanistic assumptions are now published doesn't mean they are now definitive 'proof'.

Response: Thanks for the reviewer's suggestions. It is indeed more appropriate to indicate that previous studies have "shown" or "demonstrated" a specific phenomenon rather than claiming definitive "proof." We have revised "prove" to "shown"/"demonstrated". We have replaced all "prove" throughout the revised manuscript.

Regarding the literature supporting the role of the proton motive force in facilitating the reversing function of NADH dehydrogenase, all the above-mentioned references have been included in the revised manuscript. We have summarized the relevant studies that report proton motive force driven NADH regeneration, categorizing them into four main groups:

- 1) In mitochondria, the reverse electron transfer, also the reversing function of NADH dehydrogenase, was observed ^[1]. This process can reduce NAD⁺ to NADH but result in reactive oxygen species (ROS) generation in the presence of oxygen, which can be toxic to cells ^[2]. Researchers found that the proton motive force is a key factor in driving the reverse electron transfer. Researchers have identified the proton motive force as a crucial factor driving reverse electron transfer to reduce NAD⁺ ^[3,4]. Leveraging this mechanism, they have employed the proton motive force to catalyse NADH generation ^[5].
- 2) In photosynthetic bacteria, transfer of reducing equivalents from the quinone to NAD⁺ by reverse flow at NADH dehydrogenase is the key process to generate reducing power ^[6]. As we mentioned in Point 2, *R. palustris* is a photosynthetic bacterium, as a model system for studying extracellular electron uptake. Relevant studies ^[7,8] show that the proton motive force is important to drive its electron transfer and NADH generation.

- 3) In chemolithotrophs, similar to photosynthetic bacteria, they obtain the proton motive force by their respiration pathway^[9]. For example, they use nitrate or sulfate as the electron acceptor, leading to the generation of a proton motive force that drives the production of NADH^[10].
- 4) In *Shewanella oneidensis*, the Mtr-mediated inward electron transfer was identified, allowing electrons to be transferred from the electrode to the quinol pool^[11]. For further NADH generation, the importance of proton motive force has been reported to drive the reverse electron transfer^[12, 13].

[1] Chance, Britton, Hollunger, Gunnar. Energy-Linked Reduction of Mitochondrial Pyridine Nucleotide. *Nature* 185, 666–672 (1960).

[2] Scialò, Filippo, Daniel J. Fernández-Ayala, and Alberto Sanz. "Role of mitochondrial reverse electron transport in ROS signaling: potential roles in health and disease." *Frontiers in physiology* 8 (2017): 428.

[3] Chance, Britton, and Gunnar Hollunger. "The interaction of energy and electron transfer reactions in mitochondria: I. General properties and nature of the products of succinate-linked reduction of pyridine nucleotide." *Journal of Biological Chemistry* 236.5 (1961): 1534-1543.

[4] Chance, Britton, Howard Lees, and John R. Postgate. "The meaning of "reversed electron flow" and "high energy electron" in biochemistry." *Nature* 238.5363 (1972): 330-331.

[5] Wright, John J., et al. "Reverse electron transfer by respiratory complex I catalysed in a modular proteoliposome system." *Journal of the American Chemical Society* 144.15 (2022): 6791-6801.

[6] Dupuis, A., et al. "The complex I from *Rhodobacter capsulatus*." *Biochimica et Biophysica Acta (BBA)-Bioenergetics* 1364.2 (1998): 147-165.

[7] Guzman, Michael S., et al. "Phototrophic extracellular electron uptake is linked to carbon dioxide fixation in the bacterium *Rhodopseudomonas palustris*." *Nature communications* 10.1 (2019): 1355.

[8] Liu, Xing, et al. "Syntrophic interspecies electron transfer drives carbon fixation and growth by *Rhodopseudomonas palustris* under dark, anoxic conditions." *Science Advances* 7.27 (2021): eabh1852.

[9] Kiesow, Lutz A. "Energy-linked reactions in chemoautotrophic organisms." *Current topics in bioenergetics*. Vol. 2. Elsevier, 1967. 195-233.

[10] Peck Jr, H. D. "Energy-coupling mechanisms in chemolithotrophic bacteria." *Annual Reviews in Microbiology* 22.1 (1968): 489-518.

[11] Ross, Daniel E., et al. "Towards electrosynthesis in *Shewanella*: energetics of reversing the Mtr pathway for reductive metabolism." *PloS one* 6.2 (2011): e16649.

[12] Rowe, Annette R., et al. "Tracking electron uptake from a cathode into *Shewanella* cells: implications for energy acquisition from solid-substrate electron donors." *MBio* 9.1 (2018): 10-1128.

[13] Tefft, Nicholas M., Kathryn Ford, and Michaela A. TerAvest. "NADH dehydrogenases drive inward electron transfer in *Shewanella oneidensis* MR-1." *Microbial Biotechnology* 16.3 (2023): 560-568.

Minor issues.

4. Can the authors please specify why expression of this MtrCAB/Rhodopsin in R.

eutropha was novel or even advantageous when compared to the previous study in *Shewanella oneidensis* (reference #9)

Response: A key goal of synthetic biology is to engineer organisms that can use renewable energy, such as solar energy and electricity, to convert CO₂ to biomass, chemicals, and fuels. The choice of different host microorganisms leads to different research fields. We chose *R. eutropha* and reference #9 chose *S. oneidensis*, which are two different strategies. *R. eutropha* offers a significant advantage due to its native CO₂-fixation pathway, and it is widely used as a bio-factory for producing valuable chemicals from CO₂. In addition, our lab has studied *R. eutropha* for many years and aims to construct a new energy generation system in *R. eutropha* to convert it to a photoelectrotroph. Engineering *S. oneidensis* is also a promising approach. *S. oneidensis* has a native and powerful extracellular electron transfer pathway (i.e. the Mtr-mediated pathway). We have read relevant papers from the group for reference #9. Their aim was to construct a CO₂-fixation pathway in *S. oneidensis* to make it an electroautotrophic bacterium. Our focus is more on the heterologous expression of energy modules and the expression of MtrCAB/Rhodopsin in *R. eutropha* serves as a valuable reference for future research on the modification of other chemoautotrophs.

We added description about the aim of conversion of chemoautotrophic bacteria to photoelectrotrophic bacteria in the Introduction section as follows:

“Numerous studies demonstrated that the engineered bacteria with rhodopsin are able to convert light energy into intracellular chemical energy. For example, the engineered *E. coli* with rhodopsin has been shown improved bioproduction in the light. Additionally, the expression of rhodopsin in the electroactive bacterium *S. oneidensis* can power the electrosynthesis of reduced products in cathodic conditions. Integrating a rhodopsin-based photosystem with chemoautotrophic bacteria represents a novel approach to increase biosynthesis from CO₂. Chemoautotrophic bacteria can utilise inorganic chemicals or an electrode as the electron donor to assimilate CO₂ into valuable compounds. *R. eutropha* H16, a model chemolithoautotroph, is a promising microbial chassis, owing to its CO₂ fixation pathway and metabolic versatility. We recently engineered *R. eutropha* H16 with a *Gloeobacter* rhodopsin (GR) and created a redox loop by integrating it with an external electrode.....”.

5. Line 86; can the authors provide quantitative values for the improvements of ET rates and proton pumping capacity? If not, can the authors quantitatively reveal this in terms of improved growth, not just growth rate, but overall yield?

Response: Thanks for the reviewer's suggestions to quantify values for the improvement of electron transfer rate and proton pumping capacity. While it may be challenging to directly quantify the specific electron transfer rate, we can assess the electron transfer over the course of 5 days by measuring the total charge passing through the electrodes. Our calculations indicate that the total charge passed through the electrodes with *R. eutropha*-GR-Mtr increased by 54.4% compared to the photoelectrochemical system with *R. eutropha*-GR. We have added this to the results section with the following sentence: “....., the total charge transfer of the photoelectrochemical system with *R. eutropha* RHM5-GR-Mtr increased by 54.4% compared to that with *R. eutropha* RHM5-GR.” In addition, the faradaic efficiency could reflect the yield because it indicated the energy of biomass produced with a specific charge input. The faradic efficiency of *R. eutropha* RHM5-GR-Mtr increased to 35.4% from 23.9%, compared to *R. eutropha* RHM5-GR. Based on the faradic efficiency, we can stoichiometrically deduce that their biomass yield was ~0.62 mmol of produced biomass per 1 mol electrons fed to *R. eutropha* RHM5-GR and ~0.92 mmol of produced biomass per 1mol electrons fed *R. eutropha* RHM5-GR-Mtr. We

have also added this to the revised manuscript. As for the improvement of proton pumping capacity, we have added the additional experiments to compare the pH change to reflect the increase of proton pumping capacity resulted from the addition of canthaxanthin. “We resuspended the *R. eutropha* RHM5-GR strains with or without canthaxanthin in an unbuffered solution and tracked the extracellular proton concentration changes over 1 min in the light. The presence of canthaxanthin resulted in a nearly double the ΔpH of compared to the strain with GR alone.” The new data has been added in supplementary information as fig. 9.

Supplementary fig. 9. Extracellular proton concentration changes in cells suspensions with GR and GRCX over 1 min. Statistics were performed with Student's t-test (*: $p < 0.05$, data are means \pm SD, $n = 3$).

6. Line 234: The authors state 'canthaxanthin' can 'significantly enhance' photosynthesis. But I would argue that the difference, while statistically 'significant', is very minor (just a few %). Please write the quantitative % increase instead.

Response: Thank you for your suggestions. We have added the quantitative % increase to the revised manuscript “In the presence of canthaxanthin, the doubling time of the photo-electrotrophic growth reduced by 22% to 70 h (Fig. 4g) and the efficiency increased by 21% to 42.9% (Fig. 4h) compared to the group with GR only,

7. Line 247: a ~2.7 hour increase in doubling time (from 70h without to 67.3h with the addition of carbonic anhydrase) is minor in my opinion. Furthermore, batch to batch variability is not reported, what is the n-value here? I do not see it in the Figure 4 caption, in which it should be added.

Response: We conducted experiments in triplicate for each engineered strain under different conditions. The figure caption now includes 'n=3' to indicate the number of replicates. The data has been summarised in the table below, and statistical analysis was performed using Student's t-test, yielding a p-value of 0.046. Source data are provided with this paper.

	GRCX-Mtr	GRCX-Mtr- can	Student's t test
Replicate 1	69.16 h	67.98 h	$p=0.046$
Replicate 2	70.15 h	65.83 h	
Replicate 3	71.08 h	68.34 h	

8. Line 56- poor grammar, the sentence is difficult to understand.

Response: Thank you for pointing that out. We have revised the sentence to “Numerous studies demonstrated that the engineered bacteria with rhodopsin are able to convert light energy into intracellular chemical energy.”

9. Abstract: I find it inconsistent that the authors list the species name for the host strain, the rhodopsin (that was previously incorporated), but omit the species name of the newly introduced MtrCAB electron transfer conduit in the abstract (which is their primary novelty in this work).

Response: We have added descriptions “...an extracellular electron transfer pathway of *Shewanella oneidensis* MR-1...” and “...MtrCAB from *S. oneidensis*...” in the revised abstract.

10. Lack of clarity: From the abstract alone, it was not easy for me to pinpoint what innovation the authors made, without diving into their previous work. Only upon reading the following in the intro: ‘In this study, *R. eutropha* H16 was engineered to heterogeneously express the MtrCAB 73 complex and GR protein’ could I begin to identify their novelty. Due to the extensive engineering of this strain, the authors should clarify what was incorporated in the strain previously (perhaps already in the abstract), and what specific novelty was added in this work (the MtrCAB).

Response: Thanks for the reviewer’s suggestions. The novelty of the study is the heterologous expression of Mtr pathway and its coupling with rhodopsin-powered proton pumping establishing an innovative way for proton-coupled electron transfer to fuel CO₂ fixation. In the revised abstract, we emphasised the importance of the combination of electron transfer uptake based on the Mtr pathway from *Shewanella oneidensis* and the GR-driven proton movement, as follows: “.....Here, we constructed an artificial photosynthesis system which combined the proton-pumping ability of rhodopsin with an extracellular electron uptake mechanism, establishing a novel pathway to drive photoelectrosynthetic CO₂ fixation by *Ralstonia eutropha* (*Cupriavidus necator*) H16, a facultatively chemolithoautotrophic soil bacterium. *R. eutropha* was engineered to heterologously express an extracellular electron transfer pathway of *Shewanella oneidensis* MR-1 and *Gloeobacter* rhodopsin (GR). Employing GR and the outer-membrane conduit MtrCAB from *S. oneidensis*, extracellular electrons and GR-driven proton motive force were integrated into *R. eutropha*’s native electron transport chain (ETC).....”

11. General comments: The authors increased GR’s light-harvesting capacity using a dye (canthaxanthin). I am curious if the authors have thought of doing a simple power study, in which growth under increased (or decreased) LED intensity was monitored. A conclusion on where light harvesting becomes rate limiting could be reached using such a method. Off-target affects of the dye, e.g. local heating, are not discussed.

Response: Thanks for the reviewer’s suggestion to improve the study. The light intensity applied, set at ~150 μmol/s/m², was based on instrumental limitations to ensure consistent and stable light exposure. We have investigated the effect of the decreased LED intensity (0, ~50 and ~100 μmol/s/m²) on biomass growth (Supplementary fig. 12). The new data has been added in supplementary information as supplementary fig. 12.

Supplementary fig. 12. Biomass increases in *R. eutropha* RHM5-GR-Mtr calculated from changes in the OD₆₀₀ over 5 days under different light intensities. Statistics were performed with Student's t-test (*: $p < 0.05$, **: $p < 0.01$, ***: $p < 0.001$, ****: $p < 0.0001$, data are means \pm SD, $n = 3$).

To address the concern about the off-target affects of the dye, we conducted a simple experiment to compare the viability of the cells with GR and GR-canthaxanthin under illumination. In the experiment, we grew the GR-expressing cells, with and without canthaxanthin, respectively. Then we harvested them, washed them with phosphate buffer solution (PBS) twice, and resuspended them in PBS to an OD of 0.5. After 72-hour incubation in light, we observed that there was not significant decrease in cell viability in the group containing canthaxanthin compared to the group without canthaxanthin, indicating that the canthaxanthin did not adversely affect the system performance. In addition, the GR-canthaxanthin has been reported to have a higher thermal stability compared to GR [1]. The new data has been added in supplementary information as supplementary fig. 13.

Supplementary fig. 13. Comparison of the viability of the engineered cells with and without canthaxanthin after 3-day incubation under illumination. (n.s.: not significant, data are means \pm SD, n = 3).

We have added the discussion to the revised manuscript as follows: “In this system, the proton motive force primarily relies on the photoreaction of the proton-pumping rhodopsin. There are two strategies for enhancing the proton pumping of rhodopsin. One approach is increasing light intensities, thereby elevating the proton pumping rate of the rhodopsin. We found that increasing light intensities led to enhanced biomass increases in *R. eutropha* RHM5-GR-Mtr (Supplementary fig. 12). The other approach is binding an antenna molecule to rhodopsins. In nature, microorganisms cannot control the light intensities of natural light, but they synthesise the antenna molecules to bind with rhodopsin for enhanced energy absorption and energy generation. In our system, we introduced the exogenous antenna molecule canthaxanthin which can be potentially biosynthesised. The addition of canthaxanthin did not adversely affect the viability of cells (Supplementary fig. 13). Furthermore, the GR-canthaxanthin complex was reported to have higher thermal stability compared to GR.”

[1] Chuon, K. et al. The role of carotenoids in proton-pumping rhodopsin as a primitive solar energy conversion system. *Journal of Photochemistry and Photobiology B: Biology* 221, 112241 (2021).

Reviewers' Comments:

Reviewer #1:

Remarks to the Author:

The authors of this manuscript (NCOMMS-23-29522A) have provided a proper, high-quality, response to all my comments (i.e., both major and minor), except for two of them:

1] The least important of these two is the issue of whether or not it has sufficiently been demonstrated that the heterologously expressed MtrABC system is indeed involved in electron transfer from the electrodes to the cells. The authors have added a figure (Fig. S5), to provide further evidence supporting this. Although I do agree that this figure is providing further evidence, there still is the remote possibility that cloning of the *mtr* system increases the permeability of the bacterial cell envelope (i.e. CM and OM) for endogenous redox intermediates like riboflavins and quinone biosynthetic intermediates. But I could agree with the conclusion that this indeed is a remote possibility.

2] The second, more important point is the calculation of the efficiency of conversion of (solar) light into biomass. For the equation as used, it would require that the authors would have illuminated their 'hybrid photosynthesis system' through a PV cell, which they very likely have not done (at least Fig. 1 suggest they did not do this) for the measurement of biomass yield. -- I hope the authors will agree and reconsider their statement on the overall efficiency of this 'hybrid photosynthesis system'.

Reviewer #2:

Remarks to the Author:

All of my comments have been addressed.

Reviewer #3:

Remarks to the Author:

The results on the electron transfer from the Mtr are not entirely convincing when considered individually (the SDS-PAGE expression band is very weak (at best); the C-D Raman signal is noisy; the 54.4% enhancement in charge transfer is not shown). The authors calculated an increase of 54.4% in the amount of charge passed in the Mtr strain vs. the non-Mtr strain, but I could not find that revised text in the manuscript. It is also unclear how this value was calculated.

Considering all these arguments together, the authors could make the case that the Mtr may be helping, though the performance seems very limited by the expression level based on the SDS-PAGE. I would recommend to at least include a comment on this performance and possible ideas for future work on improving the electron transfer efficiency (improving expression or through expression of additional protein), which seems to be limiting.

Reviewer #4:

Remarks to the Author:

I'm satisfied with authors' thorough revision of the manuscript, feeling they have satisfactorily answered all reviewer remarks and improved the quality of the manuscript to the standards for publication in Nat Comm.

REVIEWER COMMENTS

Reviewer #1 (Remarks to the Author):

The authors of this manuscript (NCOMMS-23-29522A) have provided a proper, high-quality, response to all my comments (i.e., both major and minor), except for two of them:

1] The least important of these two is the issue of whether or not it has sufficiently been demonstrated that the heterologously expressed MtrABC system is indeed involved in electron transfer from the electrodes to the cells. The authors have added a figure (Fig. S5), to provide further evidence supporting this. Although I do agree that this figure is providing further evidence, there still is the remote possibility that cloning of the mtr system increases the permeability of the bacterial cell envelope (i.e. CM and OM) for endogenous redox intermediates like riboflavins and quinone biosynthetic intermediates. But I could agree with the conclusion that this indeed is a remote possibility.

Response: We appreciate the reviewer's feedback. The expression of MtrCAB system could increase the permeability of the outer membrane for the external redox mediators. The electron mediators passing through the outer membrane could be involved in the electron transfer and enhance the overall electron transfer rate. We have added the discussion to the revised manuscript, as follows:

".....In addition to Mtr-mediated electron transfer, the expression of MtrCAB complex could also increase the permeability of the outer membrane, facilitating the entry of the external redox mediator, flavins into the cells for electron transfer. These flavin molecules play an important role in periplasmic electron transfer and can deliver electrons to the redox enzymes of the electron transport chain, thereby further increasing the electron transfer rate."

2] The second, more important point is the calculation of the efficiency of conversion of (solar) light into biomass. For the equation as used, it would require that the authors would have illuminated their 'hybrid photosynthesis system' through a PV cell, which they very likely have not done (at least Fig. 1 suggest they did not do this) for the measurement of biomass yield. -- I hope the authors will agree and reconsider their statement on the overall efficiency of this 'hybrid photosynthesis system'.

Response: We have carefully reconsidered our statement on the efficiency. To make it clear, we have revised the Fig. 1 and restated the efficiency " η " as "**energy transfer efficiency**" rather than "photosynthesis efficiency" or "solar-to-biomass conversion efficiency" by adding more detailed descriptions. We believe "energy transfer efficiency" can more accurately describe the essence of the equation. Please see the follows for the details.

Figure 1 revision

The original Fig. 1 did not effectively illustrate the hybrid system, potentially causing misleading and confusion about the calculation of efficiency. Therefore, we have revised Fig. 1 to indicate that light is directed both to the PV cell and the reactor:

Fig. 1 Construction of hybrid photosynthesis by combining an electrochemical system with engineered cells expressing rhodopsin and an outer-membrane conduit Mtr.

Equation clarification and assumptions explained:

We compared our calculation to other reports of artificial photosynthesis systems [1,2], their efficiency calculations are based on the equation:

$$\eta = \frac{\Delta_r G^\circ \text{ gain from CO}_2 \text{ to biomass} \text{ ----Output}}{\text{Charge passed (C)} \times \text{Voltage (V)} / \eta_{\text{solar-to-electricity}} \text{ ----Input}}$$

In their systems, the energy input is based on the light energy absorbed by the PV cell, which serves as the only light-absorbing component. In contrast, our system comprises two valid energy-absorbing components: the PV cell and the bacterial rhodopsin. Therefore, we calculated the light energy absorbed by each component separately and subsequently combined them as the total light energy input.

We used the following equation to calculate the theoretical efficiency:

$$\eta = \frac{\Delta_r G^\circ \text{ gain from CO}_2 \text{ to biomass}}{\text{Charge passed (C)} \times \text{Voltage (V)} / 0.2 + \text{proton pumped by rhodopsin} \times 205000(\text{J}) / 0.6}$$

The energy input is the valid energy absorbed by the hybrid system, i.e., $E_{\text{input}} = E_{\text{solar panel}} + E_{\text{rhodopsin}}$. Based on the revised Fig. 1, we calculated the energy efficiency by making the following assumption:

1. Light energy is evenly distributed as beams, with some directed at the PV cell and some at the reactor.
2. Light energy is simultaneously utilised by two energy-absorbing components: the PV cell and the rhodopsin of bacteria, and they use light energy independently.
3. The electricity generated by the PV cell is calculated as 'Charge passed (C) × Voltage (V),' and the energy absorbed by rhodopsin is calculated by 'proton pumped by rhodopsin × photon energy.'
4. We assume the light-to-electricity efficiency is 0.2 and the quantum efficiency (photon-to-proton) of the rhodopsin is 0.6.

As a result, the theoretical input energy can be expressed as 'Charge passed (C) × Voltage (V)/0.2 + proton pumped by rhodopsin × 205000(J)/0.6.' These assumptions have been added to our Materials and Methods section.

Our calculation is to reflect the energy dynamics within the hybrid system; thus we think “energy transfer efficiency” can accurately describe the essence of the equation.

We have also revised the caption of the Supplementary fig. 15 to make it as “Energy transfer efficiency estimation”.

Input Light Energy : Electrical Energy/0.2 + Proton motive force/0.6

Electrical Energy : Charge passed through (C)× Applied voltage (V)

Proton motive force : proton pumping capacity (mmol H⁺/gCDW/h)× ΔG (J/mol) × ∫μ×X dt

[μ= Specific growth rate (h⁻¹); X = biomass (gCDW); t= time (h)]

Output: Biomass Energy= Biomass (mol))× ΔG (J/mol)

Supplementary fig. 15 Energy transfer efficiency estimation by assuming solar-to-electricity efficiency of 0.2 and quantum efficiency of rhodopsin of 0.6.

[1] Liu, Chong, et al. "Water splitting–biosynthetic system with CO₂ reduction efficiencies exceeding photosynthesis." *Science* 352.6290 (2016): 1210-1213.

[2] Torella, Joseph P., et al. "Efficient solar-to-fuels production from a hybrid microbial–water-splitting catalyst system." *Proceedings of the National Academy of Sciences* 112.8 (2015): 2337-2342.

Reviewer #2 (Remarks to the Author):

All of my comments have been addressed.

Response: We appreciate the reviewer's review and feedback.

Reviewer #3 (Remarks to the Author):

The results on the electron transfer from the Mtr are not entirely convincing when considered individually (the SDS-PAGE expression band is very weak (at best); the C-D Raman signal is noisy; the 54.4% enhancement in charge transfer is not shown). The authors calculated an increase of 54.4% in the amount of charge passed in the Mtr strain vs. the non-Mtr strain, but I could not find that revised text in the manuscript. It is also unclear how this value was calculated.

Response: We thank the reviewer's comments. We acknowledge that the band observed on the SDS-PAGE expression is relatively weak. This may be attributed to the limited expression of Mtr, and we also consider that our protein extraction method without ultracentrifugation could result in a low membrane content, as Mtr is a membrane-bound complex. Despite the band's weakness, the qualitative analysis by SDS-PAGE can help confirm the expression of Mtr. Furthermore, it's worth mentioning that Raman C-D band is less prone to a high noise level, as it is calculated by the integration of the broad range of C-D band (~150 cm⁻¹ bandwidth), instead of a single peak. The experiment was done with 294 single cells and the calculation of C-D bands have been tested by the statistical significance analysis (t test, p <0.0001, Fig. 3b).

We thank the reviewer for spotting our oversight. We compared the amount of charge passed in the Mtr strain with that of the non-Mtr strain and observed a 54.4% increase from 68 coulombs for the non-Mtr strain to 105 coulombs for the Mtr strains over 5 days. We have added it to the newly revised manuscript, as follows: "The total charge transfer of the photoelectrochemical system with *R. eutropha* RHM5-GR-Mtr increased to 105 coulombs, a 54.4% improvement compared to 68 coulombs achieved by the system with *R. eutropha* RHM5-GR."

Considering all these arguments together, the authors could make the case that the Mtr may be helping, though the performance seems very limited by the expression level based on the SDS-PAGE. I would recommend to at least include a comment on this performance and possible ideas for future work on improving the electron transfer efficiency (improving expression or through expression of additional protein), which seems to be limiting.

Response: We thank the reviewer's comments. We agree that further efforts are needed to improve the electron transfer efficiency. Although MtrCAB expression has shown promise in improving extracellular electron transfer in *R. eutropha*, the electron transfer is still limited which could be due to the low expression of MtrCAB and the lack of expression of other relevant proteins. Further work aimed at optimising the electron transfer efficiency may include:

1. Optimisation of the Mtr pathway expression. In our study, the Mtr-expressing *R. eutropha*, in the presence of flavin, showed a 1.5-fold increase in electron transfer efficiency (from 23.9% to 35.4%) compared to the non-Mtr *R. eutropha*. However, the electron transfer remains the limiting factor. The heterologous expression of MtrCAB in *E. coli* has resulted in an approximate 2 to 6-fold increase in electron transfer [1, 2]. Our observed limitation in electron transfer efficiency, compared to that reported in *E. coli*, may be due to relatively low MtrCAB expression in *R. eutropha*, according to the SDS-PAGE results. Thus, in the revised Discussion, we propose two potential strategies for enhancing Mtr expression: i) codon optimisation of *mtr* genes and ribosome binding site optimisation and ii) the addition of chemicals favourable for heme synthesis like 5-aminolevulinic acid [3].

2. Additional protein expression. Introducing additional proteins like periplasmic small tetraheme cytochrome (STC) has been reported to further enhance the

extracellular electron transfer of heterologous microorganisms [3]. This strategy has also been added to the discussion.

We have included the above discussion in the revised manuscript as follows: "This study demonstrates that the expression of MtrCAB complex enhances extracellular electron transfer in *R. eutropha*. Nevertheless, the enhancement in electron transfer remains limited compared to other heterologous microorganisms, such as *E. coli* which achieved an approximate 2 to 6-fold increase in the electron transfer efficiency through the MtrCAB expression [1, 2]. This observed limitation could be attributed to relatively low MtrCAB expression in *R. eutropha*. The expression of the Mtr pathway can potentially be improved by optimisation of codon and ribosome binding site, as well as the addition of precursors like 5-aminolevulinic acid for heme synthesis. Additionally, the introduction of additional proteins such as periplasmic small tetraheme cytochrome (STC) may further enhance electron transfer within the periplasmic space."

In addition, reviewer 1 proposed that the expression of the MtrCAB complex could increase the permeability of the outer membrane, facilitating the entry of the external electron mediator, flavins, into the cells. The flavin molecules can transfer electrons with redox enzymes in the periplasm and on the inner membrane, elevating the electron transfer rate. We have also added this discussion to the revised manuscript.

[1] Jensen, Heather M., et al. "Engineering of a synthetic electron conduit in living cells." *Proceedings of the National Academy of Sciences* 107.45 (2010): 19213-19218.

[2] Feng, Jiao, et al. "Direct electron uptake from a cathode using the inward Mtr pathway in *Escherichia coli*." *Bioelectrochemistry* 134 (2020): 107498.

[3] Mouhib, Mohammed, et al. "Extracellular electron transfer pathways to enhance the electroactivity of modified *Escherichia coli*." *Joule* 7.9 (2023): 2092-2106.

Reviewer #4 (Remarks to the Author):

I'm satisfied the with authors' thorough revision of the manuscript, feeling they have satisfactorily answered all reviewer remarks and improved the quality of the manuscript to the standards for publication in Nat Comm.

Response: We appreciate the reviewer's recognition.

Reviewers' Comments:

Reviewer #1:

Remarks to the Author:

1] Increased outer membrane permeability:

Higher outer membrane permeability by engineering the Mtr system: If the authors would be fair to my remark, I think it would be correct to leave out the word 'further' in their last sentence on this issue: "These flavin molecules play an important role in periplasmic electron transfer and can deliver electrons to the redox enzymes of the electron transport chain, thereby further increasing the electron transfer rate."

2] Efficiency:

A] Ref 46 gives a quantum yield (i.e. that is the theoretical maximum efficiency, obtained under conditions without a load!!) of 0.6 for VISUAL rhodopsin, but 0.33 for bacteriorhodopsin. In the literature the issue of QY for bacteriorhodopsin has been controversial for many years, but the consensus observation is according to my estimate that it is somewhere between 0.2 and 0.3. – The manuscript should be corrected accordingly.

B] In their revised drawing the double use of sunlight is now more clearly explained, but it is not yet properly taken into account in the calculations, in which the two contributions are added up, while – if one has real-life, large-scale, applications in mind – they each should be divided by two, because they (i.e. the PV cell and the bioreactor) will have to be placed next to each other, rather than on top of each other (i.e. only half a sun is available for each of the two processes).

Furthermore, the calculation on the newly developed system takes all assumptions to the most positive case, i.e. calculates highest efficiencies theoretically possible, but then for plant photosynthesis they are much more pessimistic by quoting 1% efficiency for a crop. It would be proper here to also use textbook efficiencies of 4.5 % for C(3) plants and 6.5 % for C(4) plants.

Reviewer #3:

Remarks to the Author:

The added details on the calculation and discussion on the electron transfer mechanism and ideas for future improvements solidified the study. I recommend publication without further revision.

REVIEWER COMMENTS

Reviewer #1 (Remarks to the Author):

1] Increased outer membrane permeability:

Higher outer membrane permeability by engineering the Mtr system: If the authors would be fair to my remark, I think it would be correct to leave out the word 'further' in their last sentence on this issue: "These flavin molecules play an important role in periplasmic electron transfer and can deliver electrons to the redox enzymes of the electron transport chain, thereby further increasing the electron transfer rate."

Response: Thank you for your suggestion. We have removed "further" from the sentence as follows: "These flavin molecules play an important role in periplasmic electron transfer and can deliver electrons to the redox enzymes of the electron transport chain, thereby increasing the electron transfer rate."

2] Efficiency:

A] Ref 46 gives a quantum yield (i.e. that is the theoretical maximum efficiency, obtained under conditions without a load!!) of 0.6 for VISUAL rhodopsin, but 0.33 for bacteriorhodopsin. In the literature the issue of QY for bacteriorhodopsin has been controversial for many years, but the consensus observation is according to my estimate that it is somewhere between 0.2 and 0.3. – The manuscript should be corrected accordingly.

Response: The reviewer is correct. The previous research on the quantum efficiency of rhodopsin has produced a wide range of values, generally falling within the range of 0.25 to 0.79 [1,2,3,4]. In order to estimate the potential of our system, we have chosen to consider both 0.25 (as minimum) and 0.79 (as maximum) in our efficiency estimation and revised the manuscript accordingly (see later in this section).

B] In their revised drawing the double use of sunlight is now more clearly explained, but it is not yet properly taken into account in the calculations, in which the two contributions are added up, while – if one has real-life, large-scale, applications in mind – they each should be divided by two, because they (i.e. the PV cell and the bioreactor) will have to be placed next to each other, rather than on top of each other (i.e. only half a sun is available for each of the two processes).

Response: We thank the reviewer's feedback. We'd like to provide clarification regarding our formula. There are two methods [5,6,7] for calculating the efficiency of light-to-biomass (η), which have been reported in artificial photosynthesis driven by PV cell:

The first [5,6] is

$$\eta = \frac{\Delta_r G^\circ \text{ gain from CO}_2 \text{ to biomass}}{\text{Charge passed (C)} \times \text{Voltage (V)} / \eta_{\text{solar-to-electricity}}} \quad (\text{Equation 1})$$

The second [7] is

$$\eta = \frac{\Delta_r G^\circ \text{ gain from CO}_2 \text{ to biomass}}{\text{Light intensity (mW/cm}^2\text{)} \times \text{Area (cm}^2\text{)}} \quad (\text{Equation 2})$$

In this study, we modified the **Equation 1** by taking into account the energy absorbed by rhodopsin in our calculations as **Equation 3**, since both the solar panel and rhodopsin absorb light energy. Then the denominator then becomes "Charge passed (C) \times Voltage (V) / $\eta_{\text{solar-to-electricity}}$ + photon energy absorbed by rhodopsin / $\eta_{\text{rhodopsin quantum efficiency}}$ "

$$\eta = \frac{\text{Energy output}}{\text{Energy absorbed by PV cell} + \text{Energy absorbed by rhodopsin}} \quad (\text{Equation 3})$$

The total light energy used in Equation 2 includes the light energy used for biosynthesis and redundant light energy, and the calculated efficiency is dependent to the illuminated area (or the position of bioreactor and PV cell).

However, in Equation 3 for this study, we only consider the light energy that are absorbed by the solar panel and rhodopsin and exclude the redundant light energy unused by both mechanisms. Thus, this is **independent** to the position of the solar panel and bioreactor, and the efficiency should not be divided by 2.

In this research we have used Equation 3, which has been used by the similar research published in Liu et al (Science 2016) [5] and Torella et al (PNAS 2015) [6], both using *Ralstonia eutropha* and hybrid systems. The denominator of Equation 3 is the **total input energy taken from light energy for biosynthesis, not the total light energy**. We have emphasised that we used the **energy transfer efficiency** as previous reports [5,6], not the **total light-to-biomass transfer efficiency**, because some photons will be dismissed without being used.

Our experimental results indicate that the electricity energy consumed for biomass increase is "Charge passed (C) × Voltage (V)"; thus "Charge passed (C) × Voltage (V)/0.2" is the light energy input required for the PV cell. Since the proton motive force generated by rhodopsin is also required by the system, we need to add the energy absorbed by rhodopsin. Hence, we sum the light energy absorbed by PV cell and rhodopsin without dividing them by two.

Revisions:

1. We have revised the description of the efficiency calculation as follows:

Energy transfer efficiency estimation

The energy transfer efficiency (η_{energy}) of the system could be calculated by dividing the energy output by the valid energy absorbed by the hybrid system from light energy, as shown in the following equation (Supplementary fig. 15),

$$\eta_{\text{energy}} = \frac{\Delta_r G^\circ \text{ gain from CO}_2 \text{ to biomass}}{E_{\text{solar panel}} + E_{\text{rhodopsin}}} = \frac{\text{Energy output}}{\text{Energy absorbed by solar panel} + \text{Energy absorbed by rhodopsin}}$$

$E_{\text{solar panel}}$ -Energy absorbed by the solar panel.

$E_{\text{rhodopsin}}$ -Energy absorbed by rhodopsin.

The Gibbs free energy gains ($\Delta_r G^\circ$) for biomass and the corresponding chemical reaction are listed as follows:

The energy absorbed by the solar panel ($E_{\text{solar panel}}$) can be calculated as "Charge passed (C) × Voltage (V) / $\eta_{\text{solar panel}}$ ". The solar-to-electricity efficiency ($\eta_{\text{solar panel}}$) is 0.2 [8]. The energy absorbed by the rhodopsin ($E_{\text{rhodopsin}}$) can be calculated as "proton pumped by rhodopsin × 205000(J) / $\eta_{\text{rhodopsin}}$ " and the quantum yield ($\eta_{\text{rhodopsin}}$) of rhodopsin varies from 0.25 to 0.79 [1-4]. The average energy per mole of photons is 205000 J [12]. The proton pumping rate of the rhodopsin is estimated at 4-10 mmolH⁺/g dry biomass/h in this study, assuming that the minimum ATP requirement is 1 mmol ATP/g dry biomass/h and ATP generation via ATPase has a stoichiometry

of one ATP per four protons^[9-11].

2. We have also revised the discussion to “For the estimation of the energy transfer efficiency of the system, we considered the energy absorbed by the solar panel and rhodopsin from light energy as the total energy input, and the increased biomass as the energy output. The total input energy taken by the hybrid system from light relies on the performance of its two light energy absorbers (i.e., the solar panel and rhodopsin). The solar-to-electricity energy efficiency of the solar panel is assumed as an attainable efficiency of 0.2^[8]. Assuming the range of quantum efficiency for rhodopsin of 0.25 to 0.79^[1-4] and the proton pumping rate ranging from 4 to 10 mmolH⁺/g-dry biomass/h^[9-11], the energy transfer efficiency can achieve 1.7%~4.8%.”

Furthermore, the calculation on the newly developed system takes all assumptions to the most positive case, i.e., calculates highest efficiencies theoretically possible, but then for plant photosynthesis they are much more pessimistic by quoting 1% efficiency for a crop. It would be proper here to also use textbook efficiencies of 4.5% for C(3) plants and 6.5% for C(4) plants.

Response: The reviewer is right. Since the calculation of the energy transfer efficiency in this study is different to that used to photosynthesis in plant, it is inappropriate to compare them. Hence, we removed the comparison of the energy transfer efficiency between this artificial photosynthesis and natural photosynthesis.

References

[1] Govindjee, R., S. P. Balashov, and T. G. Ebrey. "Quantum efficiency of the photochemical cycle of bacteriorhodopsin." *Biophysical journal* 58.3 (1990): 597-608.

[2] Birge, Robert R. "Nature of the primary photochemical events in rhodopsin and bacteriorhodopsin." *Biochimica et Biophysica Acta (BBA)-Bioenergetics* 1016.3 (1990): 293-327.

[3] Hubbard, Ruth, and Allen Kropf. "The action of light on rhodopsin." *Proceedings of the National Academy of Sciences* 44.2 (1958): 130-139.

[4] Yang, Xuchun, et al. "Quantum–classical simulations of rhodopsin reveal excited-state population splitting and its effects on quantum efficiency." *Nature Chemistry* 14.4 (2022): 441-449.

[5] Liu, Chong, et al. "Water splitting–biosynthetic system with CO₂ reduction efficiencies exceeding photosynthesis." *Science* 352.6290 (2016): 1210-1213.

[6] Torella, Joseph P., et al. "Efficient solar-to-fuels production from a hybrid microbial–water-splitting catalyst system." *Proceedings of the National Academy of Sciences* 112.8 (2015): 2337-2342.

[7] Cestellos-Blanco, Stefano, et al. "Photosynthetic biohybrid coculture for tandem and tunable CO₂ and N₂ fixation." *Proceedings of the National Academy of Sciences* 119.26 (2022): e2122364119.

[8] Cai, Tao, et al. "Cell-free chemoenzymatic starch synthesis from carbon dioxide." *Science* 373.6562 (2021): 1523-1527.

[9] Volpers, Michael, et al. "Integrated in Silico analysis of pathway designs for synthetic photo-electro-autotrophy." *PloS one* 11.6 (2016): e0157851.

[10] Claassens, Nico J., et al. "Potential of proton-pumping rhodopsins: engineering photosystems into microorganisms." *Trends in biotechnology* 31.11 (2013): 633-642.

[11] Ganapathy, Srividya, et al. "Modulation of spectral properties and pump activity of proteorhodopsins by retinal analogues." *Biochemical Journal* 467.2 (2015): 333-343.

Reviewer #3 (Remarks to the Author):

The added details on the calculation and discussion on the electron transfer mechanism and ideas for future improvements solidified the study. I recommend publication without further revision.

Response: We appreciate the reviewer's comments that have helped enhance the quality of our study.

Reviewers' Comments:

Reviewer #1:

Remarks to the Author:

In this rebuttal the authors have done a very good job to respond to the letter, but not to the spirit of my remarks.

For point 1 (increased outer membrane permeability): This to me is not a crucial point. (i.e. also, in the latest version of the manuscript, few readers will realize that the experimental evidence is also compatible with the possibility that the Mtr system has not yet been proven to be involved in the electron transfer pathway. This was the topic we discussed here, and it could have been expressed more explicitly.

For point 2: Proposals for large-scale sustainability applications for (proteo)rhodopsins have appeared in the peer-reviewed literature now for the past 49 years. But so far none of this has matured. I read the text of the manuscript as an advertisement that the current system may be able to change this. My estimate is that this claim makes no sense (yet) and should only be made if results are available at some scale (i.e. TRL3 or preferably even TRL4), to first substantiate this claim. This minimally would require measurements of the system, in a form in which both light-dependent parts are driven by sunlight.

A corollary of this opinion is also that all assumptions about pumping rate, etc. are not necessary; one can simply take the amount of sunlight energy input into the system. Furthermore, in the revised discussion the quantum yield of the proteorhodopsin of 0.8 is still considered, while I indicated in my previous response that 0.3 is already higher than what is measured in many studies (and is the value at 'level flow!'). Furthermore, reference is made for these numbers to papers like ref 10 of colleagues that have randomly collected numbers from the literature to make their point but are by no means authors of primary research papers on the topic.

REVIEWER COMMENTS

Reviewer #1 (Remarks to the Author):

In this rebuttal the authors have done a very good job to respond to the letter, but not to the spirit of my remarks.

For point 1 (increased outer membrane permeability): This to me is not a crucial point. (i.e. also, in the latest version of the manuscript, few readers will realize that the experimental evidence is also compatible with the possibility that the Mtr system has not yet been proven to be involved in the electron transfer pathway. This was the topic we discussed here, and it could have been expressed more explicitly.

Response: We thank the reviewer's feedback. We believe that Mtr-mediated nitrate reduction (Fig. 2c), D₂O testing (Fig. S5), and growth experiments (Fig. S8) collectively support Mtr's role in electron transfer. While considering the remote possibility of Mtr expression increasing membrane permeability, we acknowledge the need for a more explicit discussion, which we will address. We have added the discussion as follows: "The investigation of heterologous Mtr pathway expression in non-native bacteria has received considerable attention in recent years ^[1,2,3]. The introduction of Mtr pathway has improved the extracellular electron transfer of *R. eutropha* in this study (Fig. 4h). However, the exact functionality of the Mtr expression in *R. eutropha* and other heterologous bacteria requires more rigorous verification. For example, there is a remote possibility that MtrCAB may not be directly involved in electron transfer but, instead, might increase membrane permeability, enhancing electron transfer via endogenous electron mediators such as flavins and quinones."

For point 2: Proposals for large-scale sustainability applications for (proteo)rhodopsins have appeared in the peer-reviewed literature now for the past 49 years. But so far none of this has matured. I read the text of the manuscript as an advertisement that the current system may be able to change this. My estimate is that this claim makes no sense (yet) and should only be made if results are available at some scale (i.e., TRL3 or preferably even TRL4), to first substantiate this claim. This minimally would require measurements of the system, in a form in which both light-dependent parts are driven by sunlight.

A corollary of this opinion is also that all assumptions about pumping rate, etc. are not necessary; one can simply take the amount of sunlight energy input into the system. Furthermore, in the revised discussion the quantum yield of the proteorhodopsin of 0.8 is still considered, while I indicated in my previous response that 0.3 is already higher than what is measured in many studies (and is the value at 'level flow!'). Furthermore, reference is made for these numbers to papers like ref 10 of colleagues that have randomly collected numbers from the literature to make their point but are by no means authors of primary research papers on the topic.

Response: We appreciate the reviewer's scientific rigor. In the manuscript we did not discuss any large-scale applications yet, instead, we focus on the novelty of the engineered photoelectrosynthetic system to power microbial CO₂ fixation. We acknowledge that there may be some controversy for efficiency calculations. Following the Editor's advice, we have removed the discussion on energy efficiency calculations from the manuscript. We have the short text in the discussion as follows "Calculations can be made with respect to the overall efficiency of this system. But as yet, such calculations will have to be based on so many assumptions that a meaningful number is not yet attainable."

References:

[1] Jensen, Heather M., et al. "Engineering of a synthetic electron conduit in living cells." *Proceedings of the National Academy of Sciences* 107.45 (2010): 19213-19218.

[2] Feng, Jiao, et al. "Direct electron uptake from a cathode using the inward Mtr pathway in *Escherichia coli*." *Bioelectrochemistry* 134 (2020): 107498.

[3] Mouhib, Mohammed, et al. "Extracellular electron transfer pathways to enhance the electroactivity of modified *Escherichia coli*." *Joule* 7.9 (2023): 2092-2106.